

# NEWTON:
# Are Large Language Models Capable of Physical Reasoning?

**Yi Ru Wang**[†]    **Jiafei Duan**[†]    **Dieter Fox**[†‡]    **Siddhartha Srinivasa**[†]

[†]University of Washington    [‡]NVIDIA

{yiruwang, duanj1, fox, siddh}@cs.washington.edu

newtonreasoning.github.io

## Abstract

Large Language Models (LLMs), through their contextualized representations, have been empirically proven to encapsulate syntactic, semantic, word sense, and common-sense knowledge. However, there has been limited exploration of their physical reasoning abilities, specifically concerning the crucial attributes for comprehending everyday objects. To address this gap, we introduce 🌳 NEWTON, a **repository** and **benchmark** for evaluating the physics reasoning skills of LLMs. Further, to enable domain-specific adaptation of this benchmark, we present a **pipeline** to enable researchers to generate a variant of this benchmark that has been customized to the objects and attributes relevant for their application. The NEWTON repository comprises a collection of 2800 object-attribute pairs, providing the foundation for generating infinite-scale assessment templates. The NEWTON benchmark consists of 160K QA questions, curated using the NEWTON repository to investigate the physical reasoning capabilities of several mainstream language models across foundational, explicit, and implicit reasoning tasks. Through extensive empirical analysis, our results highlight the capabilities of LLMs for physical reasoning. We find that LLMs like GPT-4 demonstrate strong reasoning capabilities in scenario-based tasks but exhibit less consistency in object-attribute reasoning compared to humans (50% vs. 84%). Furthermore, the NEWTON platform demonstrates its potential for evaluating and enhancing language models, paving the way for their integration into physically grounded settings, such as robotic manipulation. Project site: https://newtonreasoning.github.io

## 1 Introduction

Natural Language Processing (NLP) has made remarkable progress using contextualized representations trained on extensive unprocessed text data

*Figure 1.* As works begin leveraging LLMs in physically grounded contexts, it is crucial to understand whether such models possess the ability to reason about everyday scenarios. **NEWTON**, a repository, pipeline, and benchmark, facilitates evaluation of various LLMs in a physical reasoning context.

(Zhao et al., 2023). As we start using LLMs in physically embodied pipelines (Driess et al., 2023; Ahn et al., 2022; Wu et al., 2023), it is crucial to comprehensively understand the extent that LLMs can perform physical reasoning. Some studies have proposed generalized frameworks to assess language model performance (Ribeiro et al., 2020; Kiela et al., 2021), while others have designed question answering and reading comprehension datasets to probe LLMs (Zellers et al., 2018; Chen et al., 2019; Rogers et al., 2023). However, few have explored the physical reasoning ability of LLMs.

Physical reasoning involves the cognitive process of comprehending and predicting the dynamics of physical systems based on observable phenomena and fundamental principles (McCloskey et al., 1983; Carey, 2000). It encompasses the capacity to make sense of the world by applying knowledge of attributes, such as brittleness, malleability, etc. By considering brittleness, for instance, we recognize the need to handle an object with caution, while malleability suggests that an object can be easily reshaped under pressure without fracturing. These abstract concepts enables reason-

ing about the response of objects to interactions or changes in the environment, as shown in Figure 1.

Nevertheless, creating an evaluation framework for physical reasoning is difficult, primarily due to the lack of paired object-attribute data. Humans possess a wealth of knowledge regarding the internal structure of objects and their interactions with the physical world. However, this knowledge is often implicitly acquired, making it difficult to explicitly represent such information. Previous studies in the field of physical reasoning, such as Bisk et al. (2020); Aroca-Ouellette et al. (2021), have focused on common-sense reasoning or small-scale validation. As a result, the need for a comprehensive and systematic assessment of physical reasoning remains an open problem.

To address this gap, we propose NEWTON, a **repository**, **pipeline**, and **benchmark** designed to evaluate the physical reasoning capability of LLMs. The NEWTON **repository** consists of labeled object-centric data, crowd-sourced for 700+ objects across 8 physical attributes. The NEWTON **pipeline** introduces a method for systematically generating infinite evaluation questions tailored to specific use cases. The NEWTON **benchmark** consists of 160K pre-generated questions of progressive difficulty, spanning tasks of foundational attribute comprehension, explicit application, and implicit scenario-based analysis. Extensive empirical findings demonstrate the unique contributions of NEWTON, revealing its usefulness for evaluating LLMs' understanding of underlying physics principles that dictate the behavior and properties of objects in everyday scenarios. Moreover, NEWTON effectively compliments the existing repertoire of reasoning benchmarks and datasets, further enhancing the potential to assess and refine the physical reasoning capabilities of LLMs.

## 2 NEWTON Repository

At the core of constructing an evaluation framework for physical reasoning lies the need for a representation that captures the essential attributes of objects. In this section, we highlight the **NEWTON repository**, including the identification and a shortlist of objects and attributes, and obtaining a set of consistent object-attribute annotations.

### 2.1 Objects and Attributes

To preserve grounding to physical objects, we leverage the mainstream 3D object datasets: Objaverse

(Deitke et al., 2023), YCB (Calli et al., 2015), Scanned Objects by Google (Downs et al., 2022), and Amazon Berkeley Objects (Collins et al., 2022). To establish a common representation of categories for the combined objects, we match the object title of each 3D object and approximate it to the nearest WordNet Synset (Miller, 1995). Object categories are then filtered for redundancy (repeated categories), ambiguity (uncommon, abstract categories), and irrelevance (non-manipulable objects), resulting in a curated collection of 700 common household manipulable objects. Using a subset of the objects (5%), iterations of pilot studies were run to fine-tune the format of human-facing questions, and identify the most comprehensive yet non-redundant set of physics attributes. Ultimately, we identified categories of malleability, elasticity, stiffness, softness, sharpness, surface smoothness, surface hardness, and brittleness.

### 2.2 Crowdsourcing Framework

In order to acquire accurate ground truth annotations for object-attribute pairs, we have devised a likert-scale annotation setup where annotators are tasked with selecting the most appropriate option from a 3-point scale to depict the given pair. The utilization of a 3-point likert scale serves two primary purposes: simplicity and representation of extremes. The objective of the interface is to collect responses that facilitate the categorization of data into distinctly opposing groups based on attribute conditions. Thus, the 3-point scale offers a straightforward decision-making process for users, minimizing complexity, and allowing for the selection of extreme options when there is a high level of certainty, while uncertain responses prompt the selection of the middle option. To gather these annotations, we have employed Label Studio Enterprise [1], to create tasks where a minimum of four annotators contribute overlapping annotations for each object-attribute pair.

### 2.3 Annotation Process

We provide an example of the annotation interface in Figure 2. Each annotation task consists of three main components: a question which specifies the object category and attribute, a set of 10 randomly-sampled images illustrating common objects within the object category, and three answer choices. The question is designed to incorporate some descrip-

---
[1]https://humansignal.com/platform/

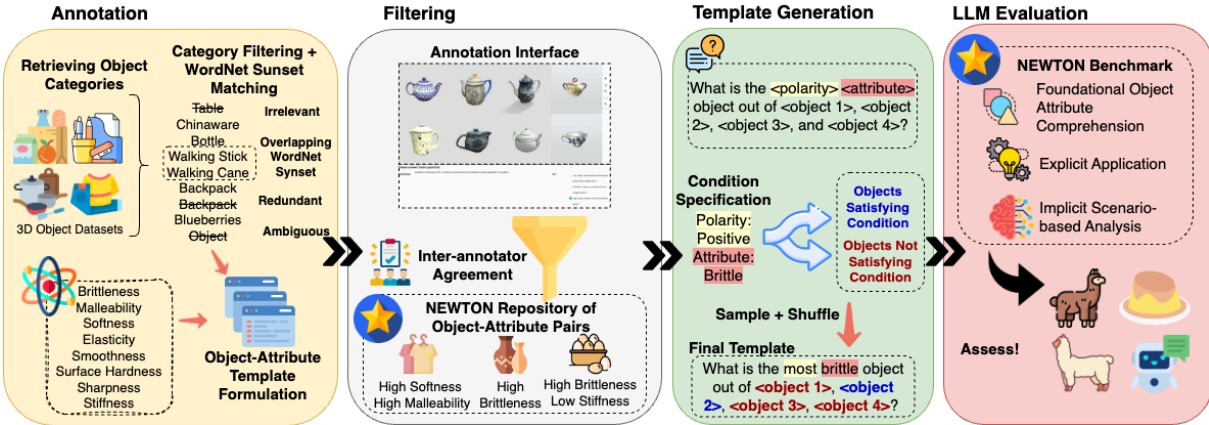

*Figure 2.* **NEWTON Pipeline.** In addition to facilitating the curation of the NEWTON repository and benchmark, the NEWTON pipeline enables convenient extensibility of evaluation measures to suit any scenario. The pipeline consists of four main components, including annotation, filtering, template generation, and LLM evaluation. The annotation component starts with retrieving object categories from 3D object datasets, the categories of which will be filtered for irrelevancy, redundancy, and ambiguity, and matched with the WordNet Synset to remove overlapping categories. We then obtain the object-attribute templates after combination with the physical attributes, and conduct the crowdsourcing process. Each object-attribute sample has a minimum of four overlapping annotations, the agreement between which is used to filter the annotations and form the NEWTON repository of object-attribute pairs. The template generation step begins with a generic template, which will be filled through condition specification and object sampling. With the generated questions, we form a benchmark of 160K questions. The pipeline also enables formulation of infinite personalized evaluation prompts to suit any intended scenario.

tion of the attribute of focus, and prompts the annotator to select an option which best describes the object, given a particular attribute topic. The annotators are asked to focus on the textual category, however, a visualization of 10 randomly-sampled thumbnails are included for reference. We provide the suite of prompts used for different attributes and potential options in Table 1. For each task, a minimum of four annotations are collected, resulting in a total of 20 000+ annotations.

## 3 NEWTON Pipeline

Leveraging the object-attribute repository, we introduce a pipeline which enables systematic generation of diverse questions based on pre-defined personalized templates, as shown in Figure 2. The process starts with crowd-sourcing human annotations on object attributes, which is then filtered based on inter-annotator agreement. The filtered data is then used to populate template-based questions in a systematic manner, a process which is extensible to a diverse range of personalized scenarios, beyond those established in the paper.

### 3.1 Annotation

Detailed analysis of the annotation interface and workflow is described in Section 2.2 and 2.3. A minimum of four annotators contributed overlapping annotations for each pair. Over 20 000 annotations were collected, featuring 700+ objects and 8 unique physical reasoning attributes.

### 3.2 Filtering

The selection of confident object-attribute pairs is contingent upon the level of agreement between annotators. To ensure reliability, a minimum of four annotations that overlap are mandated for each object-attribute pair, enabling the calculation of inter-annotator agreement. Inter-annotator agreement is calculated as a percentage of annotators who agreed on the majority response of each object-attribute task. A stringent filtering threshold of 0.75 is applied, necessitating at least three out of the four annotations to exhibit agreement. Additionally, annotations with extreme likert scores (either 1 or 3) are preserved, thereby eliminating ambiguous responses and maintaining clarity in the dataset.

### 3.3 Template Generation

Leveraging the NEWTON repository of object-attribute pairs, pre-defined templates can be filled. Each template is defined with associated object slots and attribute conditions. The condition statements are defined as $\{attribute, polarity, padding\}$, where $attribute$ defines the physics attribute used to group, $polarity$ defines whether to employ the highest or lowest extremity, and $padding$ specifies how many objects of the opposite polarity to use. Each condition statement is designed to obtain a group of $n$ objects, consisting of one object of the specified attribute and polarity, and $n-1$ objects of the negative polarity. Using the condition

| Task | Category | Template |
|---|---|---|
| Track 1: Foundational Attribute Understanding | Elasticity | If you were to deform <MASK> (by squeezing, hitting, etc.) can it recover to its original form? a) No: Object will not be able to return to its original form. b) Somewhat: Object may recover to its original form but will have signs of previous deformations. c) Yes: Object is capable of recovering near perfectly to the original form. |
| | Surface Smoothness | What is the surface smoothness of <MASK>? a) Low: majority of surface is very rough, or has many bumps. b) Moderate: some parts of surface are smooth, while some parts are not. c) High: majority of surface is smooth. |
| | Surface Hardness | How resistant to scratches or bruises is the surface of <MASK>?. a) Low: surface can be easily scratched or bruised. b) Moderate: surface is resistant to most unintentional damage, but could be scratched if force is applied. c) High: object does not get scratched or bruised easily. |
| | Brittleness | How easy is it for <MASK> to break (shatter, crack) when a sudden impact force is applied? a) Low: object can withstand most impact forces (drop, smash, etc.). b) Moderate: object can withstand minor impact forces. c) High: object shatters easily with impact force. |
| | Softness | What is the softness (fluffyness, squishiness) of <MASK>? a) Low: object is hard. b) Moderate: some parts of the object are soft. c) High: the majority of the object is soft. |
| | Sharpness | What is sharpness of <MASK>? a) Low: object does not have sharp corners/edges and is not capable of piercing. b) Moderate: object may have sharp corners / edges but is not designed to pierce. c) High: object is designed to pierce. |
| | Stiffness | What is the amount of continued weight that <MASK> can withstand before deforming or cracking? a) Low: object can support very minimal weight before deforming or cracking. b) Moderate: object can withstand a small amount of weight, but something heavy can break the object. c) High: object can easily withstand more than a 10 kg weight. |
| | Malleability | Can <MASK> be reshaped to other forms? a) No: object cannot be reshaped. b) Somewhat: object can be slightly reshaped. c) Yes: the object can be reshaped to most arbitrary forms. |
| Track 2: Explicit Application | Boolean | True/False: <object> is <attribute>. True/False: <object> is not <attribute>. |
| | Boolean | True/False: <object 1> is more <attribute> than <object 2>. True/False: <object 1> is less <attribute> than <object 2>. |
| | Multiple Choice | Which object is the most <attribute>? Options: <object1>, <object 2>, <object 3>, <object 4>. Which object is the least <attribute>? Options: <object1>, <object 2>, <object 3>, <object 4>. |
| Track 3: Implicit Scenario-based Analysis | Scenario 1 | Attribute: Stiffness (high), Brittleness (low) Context: I am packing a backpack. Question: Which of <object 1>, <object 2>, <object 3>, <object 4> should I put at the bottom? |
| | Scenario 2 | Attribute: Malleability (high), Elasticity (high) Context: I have an irregularly shaped space in my suitcase, and four objects with the same volume. Question: Which of <object 1>, <object 2>, <object 3>, <object 4> could fit in that space? |
| | Scenario 3 | Attribute: Surface Hardness (high), Surface Smoothness (low) Context: I need an object to place sandpaper above. Question: Which of <object 1>, <object 2>, <object 3>, <object 4> is the most suitable? |
| | Scenario 4 | Attribute: Sharpness (high) Context: I am trying to open some plastic packaging. Question: Which of <object 1>, <object 2>, <object 3>, <object 4> can help me open? |
| | Scenario 5 | Attribute: Softness (high), Elasticity (high) Context: I am wrapping fragile gifts and want to protect them from impact. Question: Which of <object 1>, <object 2>, <object 3>, <object 4> should I choose for cushioning? |
| | Scenario 6 | Attribute: Surface hardness (high), brittleness (low), stiffness (high) Context: I need to hammer a nail into a solid wooden board. Question: Which of <object 1>, <object 2>, <object 3>, <object 4> should I choose? |
| | Scenario 7 | Attribute: Sharpness (high) Context: I need to prepare a work table as a play area for kids. Question: Which of <object 1>, <object 2>, <object 3>, <object 4> should I remove? |
| | Scenario 8 | Attribute: Brittleness (low), Elasticity (high) Context: I have a robot which sorts objects by tossing them to bins. Question: Which of <object 1>, <object 2>, <object 3>, <object 4> should I remove? |
| | Scenario 9 | Attribute: Softness (high), Elasticity (high) Context: I need an object to provide insulation from a sharp edge on a piece of furniture. Question: Which of <object 1>, <object 2>, <object 3>, <object 4> should I choose? |

*Table 1.* **Question Templates.** Styles of questions across the three tracks in the NEWTON benchmark. There are three tracks: foundational attribute comprehension, explicit application, implicit scenario-based analysis. Track 1: Foundational attribute comprehension consist of identical questions to those used in the human-annotation process. Track 2: Explicit application consist of questions where the object attribute is mentioned explicitly in the query, which is formatted in Boolean or Multiple Choice style. Track 3: Implicit Scenario-based Analysis consists of implicit questions, where the attribute(s) of focus is not explicitly mentioned.

statements as a filter, we can obtain groups of object-attribute pairs which satisfy the given conditions and fill in the object slots, the process of which is shown in Figure 2.

# 4   NEWTON Benchmark

We introduce the NEWTON benchmark, a tool to assess the cognitive ability of language models to understand and reason about physical attributes of everyday objects. The NEWTON benchmark comprises of three progressively challenging tracks, and have a combined 160k questions covering 700+ objects and 8 unique attribute categories.

## 4.1   Tasks

NEWTON benchmark has 160k questions distributed over three reasoning tracks, namely Foun-

dational Attribute Comprehension, Explicit Application, and Implicit Scenario-Based Analysis. These tracks are selected to align with facets within Bloom's cognitive taxonomy, including comprehension, application, and analysis (Adams, 2015). The underlying task involves multiple choice question answering, where given a query **q** and up to four possible candidate choices, $c_{1...4}$ the language model must select the correct option, $c_{correct}$ of which there is exactly one for any given query.

**Foundational Attribute Comprehension.** The first step to understanding concepts in an object-centric manner is to make the connection between objects and their attributes. This one-dimensional reasoning is the core assessment strategy within the Foundational Attribute Comprehension track, where each question involves understanding a single object-attribute pair. This track serves as a means to gauge the disparities in the distribution of comprehension between humans and language models regarding object attributes. Questions in this track closely mirror those presented to human annotators, as shown in Table 1, with minimal adjustments made to accommodate the diverse prompting formats required for different models.

**Explicit Application.** To be able to apply knowledge of understanding attributes in reasoning-type tasks is crucial for language-model integration in downstream tasks. Hence, the explicit application task aims to evaluate the language model's capacity to effectively apply their understanding of object attributes in explicit reasoning tasks. Through a combination of Boolean and multiple choice questions, language models must reason about the correctness of statements concerning individual objects or pairs of objects, as shown in Table 1.

**Implicit Scenario-based Analysis.** This track assesses the language model's aptitude for reasoning in scenario-based tasks where the attribute to be inferred is not explicitly mentioned. Each prompt presented within this track has two components: context and question. The context serves as a description of the scenario, and implicitly highlights the attributes(s) of focus. The question presents candidate objects, one of which is the correct answer. We show example templates in Table 1.

## 4.2   Statistics

In total, NEWTON benchmark consists of 160K questions distributed over the three tasks of Foundational Attribute Understanding, Explicit Appli-

cation, and Implicit Scenario-Based Analysis. We provide a visualization of the distribution and analysis of the data in the Appendix. In comparison to other datasets and benchmarks which have examined the topic of Physical Reasoning, NEWTON differs in its object-centric focus, providing unmatched diversity and scale as shown in Table 2.

| Factors | PIQA | PROST | NEWTON (ours) |
|---|---|---|---|
| Object-Attribute Pairs | ✗ | ✓ | ✓ |
| Physics Attributes | N/A | 3 | 8 |
| Objects | N/A | 20 | 600+ |
| Question Style | 2 answers | 4 answers | 2-4 answers |
| Questions | 16k | 19k | 160k |
| Multi-level Evaluation | ✗ | ✗ | ✓ |

*Table 2.* **Dataset Comparison.** Comparison of NEWTON with two other benchmarks aimed at Physics Understanding and Reasoning. PIQA (Bisk et al., 2020) is a dataset aimed at physics common-sense reasoning, without a focus on object-centric attribute understanding. PROST (Aroca-Ouellette et al., 2021) tackles physical and affordance reasoning from an object-centric approach, using a small subset of objects. Our dataset examines the understanding of language models from a physical reasoning perspective, with a rich and diverse set of objects, attributes, and questions.

## 4.3 Synthesis of Additional Challenge Sets

The NEWTON Repository of Object-Attribute pairs can be used to synthesize additional challenge sets, as seen in Figure 3. The process begins with Context and Attribute Specification, where users identify a context, and relevant attributes. Next, object filtering involves using the identified attribute(s) to *automatically* filter objects into a positive set and a negative set. Using the grouped objects, one can specify query templates, and automatically populate the templates to synthesize a diverse and customized challenge set. Using this, language models can be evaluated for accuracy in the specific user-identified context to find the optimal prompting strategy and model. Examples of the generated dataset are shown in the Appendix.

## 5 Results

In this section, we evaluate the performance of state-of-the-art models on NEWTON. Specifically, we quantitatively analyze the performance of the models on the tasks of foundational attribute comprehension, explicit application, and implicit analysis. We also qualitatively examine the patterns in errors made by the models.

## 5.1 Experimental Design

**Query Templates.** We make minimal changes to the prompts of language models between evalua-

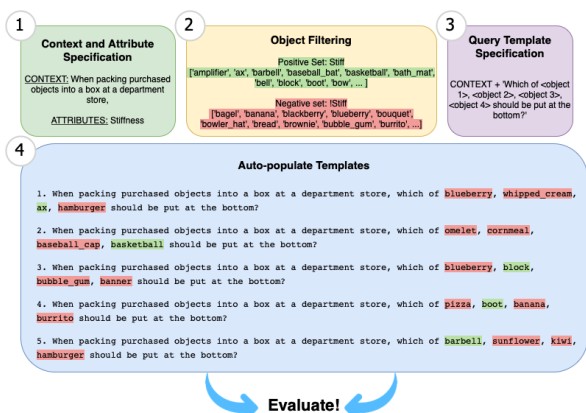

*Figure 3.* **Synthesis of additional challenge sets.** Using NEWTON's broad object-attribute combinations, users can design custom challenge sets for their needs. The process starts with Context and Attribute Specification. Objects are then *automatically* filtered based on chosen attributes into positive and negative sets. With these, users can quickly create tailored challenge templates. This helps evaluate language models in user-defined contexts, optimizing prompt strategies and model selection.

| Model | Parameters | Foundation | Data |
|---|---|---|---|
| Flan-Alpaca-GPT4-XL | 3 B | T5 | Flan, GPT4-Alpaca |
| Dolly-v2 | 6.9 B | Pythia | databricks-dolly-15k |
| Flan-T5-Small | 80 M | T5 | Flan Collection |
| Flan-T5-Base | 250 M | T5 | Flan Collection |
| Flan-T5-Large | 780 M | T5 | Flan Collection |
| Flan-T5-XL | 3 B | T5 | Flan Collection |
| Alpaca | 7-30 B | LLaMA | Alpaca Data |
| BERT-SWAG | 110.1 M - 336.2 M | BERT | SWAG Dataset |
| GPT-Turbo | 154 B | GPT-3 | – |
| GPT-4 | – | – | – |

*Table 3.* **Model Details.** We provide the details of models evaluated using NEWTON, including the name of the model, number of parameters, the underlying foundation model, and the instructed or QA datasets used to fine-tune the foundation model. Note that dashes represent undisclosed details.

tions of different models. Aside from the particular formatting requirements necessary for inference by different models, we make no changes to the format of the prompt. Prompt structures for different families of models are illustrated in the Appendix. **Models.** We provide an outline of the various models bench-marked with NEWTON in Table 3. We consider several families of large-scale pre-trained models, fine-tuned on different instructed or question answering datasets.
**Metrics.** Two metrics are used in the evaluation of LLMs' performance across the three benchmark tracks: Agreement (%) and Accuracy (%). Agreement (%) is used to evaluate Track 1, while accuracy is used to evaluate Track 2 and 3.

$$\text{Agreement} = \frac{|R_{\text{LM}} = R_{\text{HM}}|}{|R_{\text{LM}}|} \times \frac{|R_{\text{HM}} = R_{\text{H}}|}{|R_{\text{H}}|} \qquad \text{Accuracy} = \frac{|R_{\text{LM}} = R_{\text{HM}}|}{|R_{\text{LM}}|} \qquad (1)$$

Where $R_{\text{LM}}$ denotes the response from the language model, $R_{\text{HM}}$ denotes the majority human response, and $R_{\text{H}}$ denotes the human response. The

agreement metric regards the human agreement percentage as an upper limit and adjusts the calculated accuracy based on this percentage. This adjustment results in a higher emphasis on questions with substantial human agreement, while assigning comparatively less significance to questions where human annotator responses exhibit greater diversity. This metric aids in evaluating the extent to which language model responses align with those of humans. On the other hand, the accuracy metric considers the human-majority response as the definitive label and thus has a maximum attainable value of 100%. This metric gauges the proportion of responses that align with the majority-voted human response.

## 5.2 NEWTON Benchmark as a Diagnostic for Knowledge on Physical Attributes

The three-track setup of NEWTON benchmark enables analysis of Language Models' ability to comprehend, apply, and analyze physics attributes of everyday objects. Through a quantitative analysis, we draw several insights.

**Language models have a non-uniform comprehension of physics concepts of objects.** The foundational attribute understanding task requires classification of objects based on physical attributes. Through the task, as shown in Table 4, we find that there is an inconsistent performance of different models across the physical attributes encompassed by NEWTON. While GPT-4 (OpenAI, 2023) dominates in terms of overall performance, different models excel at different attributes. For instance, Flan-T5-Base (Chung et al., 2022) had strong performance across elasticity, stiffness, surface smoothness, and malleability, while Flan-T5-XL (Chung et al., 2022) excelled at softness. Qualitatively analyzing the common responses of language models, we find that typical errors occur due to one of three reasons: hallucination, conservatism, and misunderstanding. We illustrate examples of model outputs in the Appendix.

**Language models are capable of applying their knowledge of objects to comparison-based tasks.** By utilizing the templates highlighted in Table 1, we observe that language models exhibit proficiency in boolean and multiple-choice reasoning tasks for physical reasoning of object attributes, similar to the observations highlighted in Kadavath et al. (2022). Notably, GPT-4 (OpenAI, 2023) consistently performs well across both boolean and multiple-choice style questions, showcasing

its dominance in attributes such as surface hardness, softness, malleability, and sharpness. Alpaca-LoRA-7B (Hu et al., 2021) also excels in boolean questions, particularly in the areas of stiffness, surface smoothness, and brittleness. Additionally, other models like Flan-Alpaca-GPT4-XL, Flan-T5-XL (Chung et al., 2022), and GPT-3.5-Turbo demonstrate competitive results in various attributes.

**Some language models are capable of decision-making in implicit scenario-based tasks.** In Table 6, we present a quantitative evaluation of language models across nine scenario templates. The results show that GPT-4 (OpenAI, 2023) consistently outperforms other models by a significant margin in the majority of scenarios. It achieved an impressive overall average of 87.7% across the defined scenarios, compared to the average of 44.5% for all the evaluated models. However, it is important to note that GPT-4 struggles to provide consistently accurate responses in certain scenarios, particularly in Scenario 8. This indicates the need for evaluation schemes that carefully consider the specific deployment situations in which language models will be utilized.

## 5.3 Ablative Studies

In this section, we provide an analysis of NEWTON, focusing on potential ways of leveraging NEWTON to enhance model performance in a physical reasoning context, and examining the consistency of LLMs with regard to model size, question polarity, and answer positioning.

**Fine-tuning using NEWTON.** Aside from using NEWTON to create evaluation tools, it serves as a resource for fine-tuning pre-trained LMs, improving their grasp of physical concepts. We experiment with fine-tuning on Track 2 questions that explicitly address object attributes, then evaluate on Track 3 involving implicit attribute reasoning. Track 2 and Track 3 questions, detailed in Table 1, are distinct. Fine-tuning focuses on multiple-choice tasks using the base BERT model (Devlin et al., 2019), initially trained on SWAG (Zellers et al., 2018). We fine-tune with subsets of NEWTON's Track 2—5k, 10k, and 40k samples. Models are assessed on Track 3's implicit reasoning using NEWTON. Figure 4A reveals significant enhancement in language models (e.g., BERT) when NEWTON is part of pre-training, with increasing performance as fine-tuning samples rise. This

| Language Model | Agreement (%) | | | | | | | | |
|---|---|---|---|---|---|---|---|---|---|
| | Elasticity | Stiffness | Surface Smoothness | Surface Hardness | Softness | Brittleness | Malleability | Sharpness | Overall |
| Dolly-V2-7B | 23.7 | 11.7 | 4.1 | 3.5 | 1.3 | 1.3 | 8.6 | 2.5 | 7.6 |
| Flan-Alpaca-GPT4-XL | 9.3 | 1.1 | 59.7 | 19.9 | 25.1 | 6.1 | 9.5 | 7.4 | 15.2 |
| Flan-T5-small | 0.0 | 0.0 | 0.0 | 0.0 | 0.0 | 0.0 | 0.0 | 0.0 | 0.0 |
| Flan-T5-Base | 79.8 | 45.9 | 61.9 | 28.1 | 25.4 | 6.1 | 77.0 | 4.7 | 41.7 |
| Flan-T5-Large | 0.0 | 0.4 | 61.9 | 30.1 | 25.4 | 6.1 | 9.8 | 4.1 | 13.7 |
| Flan-T5-XL | 9.3 | 4.4 | 15.6 | 42.4 | 59.9 | 6.1 | 9.9 | 54.6 | 27.5 |
| UnifiedQA-V2-T5-Large | 0.0 | 0.0 | 61.9 | 30.5 | 10.2 | 6.1 | 9.5 | 23.5 | 14.3 |
| Alpaca-LoRa-7B | 9.3 | 0.0 | 44.3 | 0.6 | 14.4 | 3.9 | 9.5 | 1.9 | 10.0 |
| GPT-Turbo | 22.5 | 0.0 | 0.7 | 13.5 | 6.6 | 6.2 | 58.8 | 1.1 | 16.4 |
| GPT-4 | 51.3 | 8.0 | 14.1 | 47.1 | 59.5 | 76.5 | 44.1 | 58.4 | 49.7 |
| Average | 20.5 | 7.2 | 32.4 | 21.6 | 22.8 | 11.8 | 23.7 | 15.8 | 19.6 |
| Human | 89.1 | 78.7 | 79.2 | 78.9 | 89.4 | 82.6 | 86.8 | 90.5 | 84.4 |

*Table 4.* **Track 1: Foundational attribute comprehension** results for various language models. We report the agreement percentage, computed as a percentage of responses which agree with the majority voted human response, weighted by the inter-annotator agreement. We also provide the overall averaged agreement across language models, and also across attributes. In addition, we report the inter-annotator agreement average for the listed attributes for reference.

underscores NEWTON's potential for improving LLMs' physical reasoning through fine-tuning.

**Language Model Size and Impact on Performance.** To assess the influence of model size on performance, we focus on the Flan-T5 series and analyze the performance of small, base, large, and XL model sizes, as shown in Table 7. We observe that, in general, larger model sizes lead to improved performance for Track 2 and 3. However, for Track 1, the Flan-T5-Base model demonstrates the best performance. This inconsistency is likely attributed to the nature of the questions, as those in Track 1 are more descriptive in nature, while questions in Tracks 2 and 3 are more concise. The contrasting question styles likely account for the varying outcomes observed across different model sizes, as certain models may excel in handling longer, more detailed queries while others excel in providing responses to shorter, more focused questions.

**Polarity and Position Bias in Language Models.** We explore the impact of question polarity on model performance using NEWTON benchmark - Track 2. Prompting questions are categorized into Positive and Negative polarity, where Positive polarity questions include phrases like "is," "is more," and "is the most," while Negative polarity questions involve phrases like "is not," "is less," and "is the least" (as outlined in Table 1). Conducting a t-test reveals that GPT-4 and Dolly-V2-7B have no significant difference between the means of the different polarity groups, while for other models, there is a statistically significant difference ($p < 0.05$) between the means of the different polarities, indicating the presence of a polarity bias, as shown in Figure 4B. Additionally, we investigate the presence of bias based on answer position by grouping

questions according to the position of the ground truth answer, as shown in Figure 4C. Among the models, UnifiedQA demonstrates the lowest difference in the accuracy between highest and lowest scoring accuracy positions, as quantified by a t-statistic of 15.0 and $p < 0.0001$. On the other hand, Flan-Alpaca-GPT4-XL exhibits the largest difference between the highest and lowest position accuracy, quantified by a t-statistic of 170.1 and $p < 0.0001$. The observation that models exhibit inconsistencies in accuracy by position is also highlighted in (Aroca-Ouellette et al., 2021).

**Effect of Prompt Engineering.** We analyze the influence of prompt engineering on both GPT-Turbo and GPT-4. Table 8 illustrates our exploration of five distinct approaches to formulating prompts. These strategies are denoted as *V1*, *V2*, and *V3*, representing three different versions of prompts distinguished by their word choices. The *Ensemble* method amalgamates the outcomes from *V1*, *V2*, and *V3*, utilizing a majority voting system to determine the final answer. Meanwhile, the *Instruction* strategy supplements each prompt with a set of instructions, comprising eight example questions and corresponding answers that cover diverse attributes. Notably, these examples do not overlap with any of the tested questions. It's noteworthy that the Ensemble and Instruction-based prompt techniques enhance performance for Boolean-style questions. However, the performance remains comparable or, in some cases, even less favorable for Multiple Choice-style questions. These findings underscore the significance of identifying an appropriate prompt and crafting variations that are attuned to diverse question styles.

| Language Model | Elasticity | Stiffness | Surface Smoothness | Surface Hardness | Softness | Brittleness | Malleability | Sharpness | Overall |
|---|---|---|---|---|---|---|---|---|---|
| | | | | **Boolean (Accuracy %)** | | | | | |
| Dolly-V2-7B | 3.1 | 2.6 | 3.1 | 4.4 | 2.8 | 2.5 | 3.1 | 2.4 | 3.0 |
| Flan-Alpaca-GPT4-XL | 54.8 | 61.5 | 49.8 | 59.5 | 58.1 | 49.2 | 48.8 | 68.1 | 56.2 |
| Flan-T5-Small | 53.5 | 50.3 | 48.5 | 49.3 | 51.4 | 48.3 | 52.4 | 55.4 | 51.2 |
| Flan-T5-Base | 36.8 | 35.3 | 34.8 | 34.6 | 30.8 | 40.8 | 37.1 | 26.3 | 34.5 |
| Flan-T5-Large | 42.3 | 40.1 | 43.8 | 20.6 | 50.2 | 29.8 | 38.8 | 52.1 | 40.1 |
| Flan-T5-XL | 61.4 | 60.4 | 50.5 | 65.3 | 74.0 | 51.4 | 59.2 | 78.8 | 62.9 |
| UnifiedQA-V2-T5-Large | 46.6 | 51.7 | 49.8 | 57.4 | 61.1 | 50.6 | 49.7 | 54.5 | 52.7 |
| Alpaca-LoRa-7B | 61.0 | **68.2** | 67.9 | 63.9 | 62.3 | **69.5** | 62.7 | 66.8 | 65.2 |
| GPT-Turbo | 49.5 | 58.5 | 51.6 | 54.4 | 67.5 | 51.7 | 61.5 | 81.5 | 59.8 |
| GPT-4 | **66.6** | 61.7 | 60.8 | **66.8** | **78.0** | 66.8 | **71.5** | **82.3** | **69.6** |
| Average | 47.6 | 49.0 | 46.1 | 47.6 | 53.6 | 46.1 | 48.5 | 56.8 | 49.5 |
| | | | | **Multiple Choice (Accuracy %)** | | | | | |
| Dolly-V2-7B | 14.4 | 9.8 | 11.7 | 13.2 | 12.6 | 11.7 | 11.4 | 10.6 | 11.9 |
| Flan-Alpaca-GPT4-XL | **76.6** | **79.6** | 37.2 | 80.5 | 78.3 | 22.6 | 60.4 | 91.8 | 65.9 |
| Flan-T5-Small | 23.6 | 24.8 | 25.1 | 23.9 | 23.7 | 24.0 | 23.9 | 24.4 | 24.2 |
| Flan-T5-Base | 39.9 | 36.9 | 23.2 | 42.0 | 40.4 | 14.8 | 32.1 | 57.1 | 35.8 |
| Flan-T5-Large | 61.0 | 56.9 | 23.0 | 72.4 | 64.6 | 32.8 | 38.3 | 90.4 | 54.9 |
| Flan-T5-XL | 70.5 | 77.9 | 42.5 | 82.5 | 77.6 | 30.0 | 54.7 | 92.9 | 66.1 |
| UnifiedQA-V2-T5-Large | 61.7 | 59.8 | 28.8 | 71.3 | 63.7 | 27.4 | 50.3 | 75.1 | 54.8 |
| Alpaca-LoRa-7B | 26.2 | 27.8 | 24.1 | 26.6 | 26.3 | 24.3 | 24.6 | 31.2 | 26.4 |
| GPT-Turbo | 49.0 | 69.1 | 42.8 | 81.1 | 73.6 | 44.7 | 43.3 | 92.7 | 62.0 |
| GPT-4 | 75.4 | 73.8 | 64.6 | 84.8 | **91.7** | 67.8 | 72.8 | 98.5 | 78.7 |
| Average | 49.8 | 51.6 | 32.3 | 57.8 | 55.2 | 30.0 | 41.2 | 66.5 | 48.1 |
| | | | | **Combined (Accuracy %)** | | | | | |
| Dolly-V2-7B | 8.3 | 6.1 | 7.3 | 8.8 | 7.3 | 6.9 | 6.9 | 6.2 | 7.2 |
| Flan-Alpaca-GPT4-XL | 64.9 | **70.4** | 43.7 | 69.9 | 67.4 | 36.5 | 54.2 | 79.1 | 60.8 |
| Flan-T5-Small | 39.6 | 37.8 | 37.1 | 36.7 | 38.6 | 36.7 | 39.2 | 41.1 | 38.4 |
| Flan-T5-Base | 38.2 | 36.1 | 29.1 | 38.3 | 35.3 | 28.4 | 34.8 | 40.6 | 35.1 |
| Flan-T5-Large | 51.0 | 48.4 | 33.6 | 46.4 | 56.8 | 31.2 | 38.6 | 69.8 | 47.1 |
| Flan-T5-XL | 65.6 | 69.0 | 46.6 | 73.9 | 75.7 | 41.2 | 57.1 | 85.3 | 64.4 |
| UnifiedQA-V2-T5-Large | 53.6 | 55.6 | 39.6 | 64.3 | 62.3 | 39.5 | 50.0 | 64.1 | 53.7 |
| Alpaca-LoRa-7B | 44.9 | 48.4 | 46.5 | 45.4 | 45.7 | 48.0 | 45.0 | 50.3 | 46.8 |
| GPT-Turbo | 49.3 | 63.7 | 47.3 | 67.7 | 70.3 | 48.4 | 53.0 | 86.7 | 60.9 |
| GPT-4 | **70.7** | 67.6 | **62.7** | 75.7 | **84.3** | 67.2 | 72.1 | 89.8 | 73.9 |
| Average | 48.6 | 50.3 | 39.4 | 52.7 | 54.4 | 38.4 | 45.1 | 61.3 | 48.8 |

*Table 5.* **Track 2: Explicit application** evaluation results on various LLMs. We separate the questions into two streams, Boolean, which consists of True/False style questions, and Multiple Choice, which consist of QA style questions with four answer choices. We report the model accuracy across each stream, as well as the combined accuracy. For each stream, we report an averaged accuracy percentage across all models for each physical reasoning attribute. We also report an averaged accuracy percentage across all attributes, to gauge the overall understanding of language models across all physical reasoning attributes.

| | | | | | Scenarios | | | | | |
|---|---|---|---|---|---|---|---|---|---|---|
| Language Model | 1 | 2 | 3 | 4 | 5 | 6 | 7 | 8 | 9 | Overall |
| Dolly-V2-7B | 9.8 | 7.0 | 13.2 | 8.1 | 14.3 | 7.9 | 7.0 | 9.1 | 5.7 | 8.8 |
| Flan-Alpaca-GPT4-XL | 39.2 | 81.8 | 71.5 | 72.0 | 27.1 | 65.4 | 22.0 | 61.1 | 23.9 | 48.9 |
| Flan-T5-Small | 27.9 | 33.2 | 18.6 | 33.8 | 30.2 | 31.1 | 26.8 | 28.8 | 30.1 | 29.8 |
| Flan-T5-Base | 56.9 | 47.2 | 30.0 | 28.1 | 33.8 | 36.7 | 18.9 | 31.5 | 20.2 | 34.2 |
| Flan-T5-Large | 65.7 | 57.5 | 76.9 | 42.2 | 16.9 | 79.1 | 20.6 | 44.1 | 29.4 | 46.2 |
| Flan-T5-XL | 57.6 | 78.7 | 96.9 | 71.5 | 30.1 | 74.4 | 17.4 | **72.9** | 30.1 | 54.2 |
| UnifiedQA-V2-T5-Large | 55.6 | 70.9 | 83.4 | 66.0 | 33.8 | 63.5 | 39.5 | 68.5 | 45.1 | 55.4 |
| Alpaca-LoRa-7B | 24.4 | 24.6 | 26.2 | 24.4 | 24.1 | 26.4 | 24.3 | 28.6 | 24.5 | 24.8 |
| GPT-Turbo | 83.1 | 66.9 | 77.8 | 60.9 | 38.4 | 63.4 | 22.1 | 64.1 | 37.4 | 54.7 |
| GPT-4 | **95.7** | **92.5** | **100.0** | 77.9 | 75.2 | 96.8 | 91.8 | 50.8 | 84.2 | 87.7 |
| Average | 51.6 | 56.0 | 59.4 | 48.5 | 32.4 | 54.5 | 29.0 | 46.0 | 33.1 | 44.5 |

*Table 6.* **Implicit scenario-based analysis.** We present accuracies of language models on the nine scenario-based tasks from the NEWTON benchmark. Scenario 1 to 9 represents a range of scenario based questions, ranging from arrangement, to tool-use, to safety. We provide an average across each scenario as a measurement for the overall scenario complexity, as well as an average encompassing the overall performance of the model across the nine given scenarios.

| Model | Parameters | Track 1 | Track 2 | Track 3 |
|---|---|---|---|---|
| Flan-T5-Small | 80 M | 0.0 | 38.4 | 29.8 |
| Flan-T5-Base | 250 M | 41.7 | 35.1 | 34.2 |
| Flan-T5-Large | 780 M | 13.7 | 47.1 | 46.2 |
| Flan-T5-XL | 3 B | 27.5 | 64.4 | 54.2 |

*Table 7.* **Model Size' Impact on Performance.** We provide a comparison of different sizes of the Flan-T5 models evaluated using the different tracks of the NEWTON benchmark.

## 6 Related Works

**Evaluation of Reasoning Abilities.** In the past year, significant progress has been made in exploring Natural Language Processing (NLP) developments attributable to the advent of large language models. Prior studies like Bakhtin et al. (2019) focused on basic physical reasoning without language, works like Hong et al. (2021); Duan et al. (2021) have investigated physical reasoning in visual contexts, and datasets like (Clark et al., 2018; Kembhavi et al., 2017; Zellers et al., 2018) mainly assessed physical reasoning through physics questions for benchmarking pretrained models. Some studies have assessed model proficiency in the context of physical reasoning: PIQA (Bisk et al., 2020) tests models on physical commonsense, while PROST (Aroca-Ouellette et al., 2021) offers questions on physical reasoning concepts. In contrast, our work (NEWTON) introduces a framework for evaluating and improving large language models'

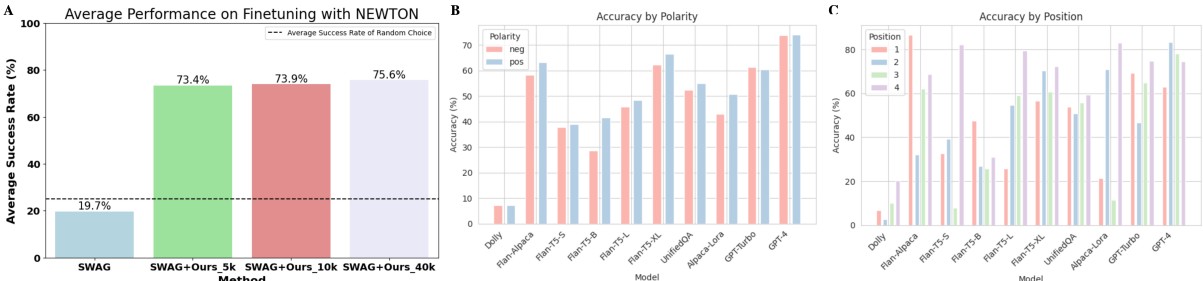

*Figure 4.* **Ablations.** From left to right: A) BERT fine-tuning results using NEWTON, note the increase in accuracy on the unseen implicit questions after finetuning on NEWTON, using of sample of 5000, 10000, and 40000, respectively. B) Accuracy by question polarity, where positive polarity represents questions phrased with is, is more, and is the most, while negative polarity represents questions phrased with is not, is less, and is the least. C) Accuracy by position, where the position value indicates the placement of the correct answer within the sequence of possible options in the question template.

| LLM + Prompting Method | E | St | SS | SH | So | B | M | Sh | Overall |
|---|---|---|---|---|---|---|---|---|---|
| **Boolean (Accuracy %)** | | | | | | | | | |
| GPT-Turbo + V1 | 51.5 | 60.5 | 53.0 | 56.0 | 71.0 | 53.5 | 57.0 | 80.5 | 60.4 |
| GPT-Turbo + V2 | 56.5 | 71.5 | 64.0 | 52.0 | 49.5 | 68.5 | 56.5 | 61.0 | 59.9 |
| GPT-Turbo + V3 | 54.0 | 53.5 | 59.0 | 60.5 | 59.0 | 61.0 | 56.0 | 60.0 | 57.9 |
| GPT-Turbo + Ensemble | 55.0 | 68.5 | 64.5 | 60.5 | 58.0 | 62.5 | 57.0 | 69.0 | 61.9 |
| GPT-Turbo + Instruction | 62.0 | 68.0 | 56.5 | 68.0 | 57.5 | 55.0 | 66.0 | 80.5 | 64.2 |
| GPT-4 + V1 | 68.5 | 67.0 | 66.5 | 65.5 | 82.5 | 63.5 | 73.0 | 86.0 | 71.6 |
| GPT-4 + V2 | 66.0 | 80.0 | 58.0 | 72.0 | 71.5 | 83.5 | 71.5 | 69.5 | 71.5 |
| GPT-4 + V3 | 75.5 | 71.5 | 61.0 | 88.5 | 73.0 | 88.0 | 72.0 | 83.5 | 76.6 |
| GPT-4 + Ensemble | 68.5 | 76.5 | 61.5 | 79.0 | 77.0 | 85.5 | 75.0 | 83.0 | 75.8 |
| GPT-4 + Instruction | 77.5 | 85.5 | 75.5 | 90.5 | 80.0 | 68.5 | 72.0 | 96.5 | 80.8 |
| **Multiple Choice (Accuracy %)** | | | | | | | | | |
| GPT-Turbo + V1 | 46.5 | 72.5 | 48.5 | 82.5 | 78.0 | 41.5 | 41.0 | 92.0 | 62.8 |
| GPT-Turbo + V2 | 25.5 | 56.0 | 36.5 | 52.5 | 55.0 | 44.5 | 38.5 | 61.5 | 46.2 |
| GPT-Turbo + V3 | 54.5 | 55.0 | 31.5 | 69.5 | 53.5 | 57.0 | 36.0 | 66.5 | 52.9 |
| GPT-Turbo + Ensemble | 36.5 | 54.5 | 33.0 | 67.0 | 56.5 | 45.0 | 34.0 | 65.0 | 48.9 |
| GPT-Turbo + Instruction | 30.5 | 53.5 | 34.0 | 63.5 | 49.5 | 32.0 | 25.0 | 69.5 | 44.7 |
| GPT-4 + V1 | 74.0 | 76.5 | 70.0 | 87.5 | 91.5 | 65.0 | 71.0 | 98.0 | 79.2 |
| GPT-4 + V2 | 49.5 | 66.5 | 37.5 | 83.5 | 70.0 | 73.5 | 69.0 | 82.0 | 66.4 |
| GPT-4 + V3 | 75.5 | 68.0 | 40.5 | 84.0 | 75.0 | 78.5 | 70.5 | 76.0 | 71.0 |
| GPT-4 + Ensemble | 65.5 | 68.5 | 47.0 | 84.5 | 81.5 | 77.0 | 71.5 | 87.5 | 72.9 |
| GPT-4 + Instruction | 62.5 | 57.5 | 54.0 | 71.5 | 65.0 | 50.5 | 31.0 | 74.0 | 58.2 |
| **Combined (Accuracy %)** | | | | | | | | | |
| GPT-Turbo + V1 | 49.0 | 66.5 | 50.7 | 69.2 | 74.5 | 47.5 | 49.0 | 86.2 | 61.6 |
| GPT-Turbo + V2 | 41.0 | 63.7 | 50.2 | 52.2 | 52.2 | 56.5 | 47.5 | 61.3 | 53.1 |
| GPT-Turbo + V3 | 54.2 | 54.2 | 45.2 | 65.0 | 56.2 | 59.0 | 46.0 | 63.2 | 55.4 |
| GPT-Turbo + Ensemble | 45.8 | 61.5 | 48.8 | 63.7 | 57.2 | 53.8 | 45.5 | 67.0 | 55.4 |
| GPT-Turbo + Instruction | 46.2 | 60.8 | 45.2 | 65.8 | 53.5 | 43.5 | 45.5 | 75.0 | 54.4 |
| GPT-4 + V1 | 71.2 | 71.8 | 68.2 | 76.5 | 87.0 | 64.2 | 72.0 | 92.0 | 75.4 |
| GPT-4 + V2 | 57.8 | 73.2 | 47.8 | 77.8 | 70.8 | 78.5 | 70.2 | 75.8 | 69.0 |
| GPT-4 + V3 | 75.5 | 69.8 | 50.7 | 86.2 | 74.0 | 83.2 | 71.2 | 79.8 | 73.8 |
| GPT-4 + Ensemble | 67.0 | 72.5 | 54.2 | 81.8 | 79.2 | 81.2 | 73.2 | 85.2 | 74.3 |
| GPT-4 + Instruction | 70.0 | 71.5 | 64.8 | 81.0 | 72.5 | 59.5 | 51.5 | 85.2 | 69.5 |

*Table 8.* **Prompt Engineering.** We tested five prompting strategies on GPT-Turbo and GPT-4: three variations (V1, V2, V3) by word choice, an Ensemble majority vote from V1-V3, and an Instruction method with eight sample Q&As. Questions were split into Boolean (True/False) and Multiple Choice types. We present accuracy for each stream and overall, as well as average accuracy per physical reasoning attribute. E, St, SS, SH, So, B, M, Sh, represents Elasticity, Stiffness, Surface Smoothness, Surface Hardness, Softness, Brittleness, Malleability, and Sharpness, respectively.

performance in physical reasoning. With 2800 object-attribute pairs and 160K QA prompts, it provides unmatched scale and extensibility, potentially setting a new standard for evaluation.

**Applications of Language Models in Physically Grounded Contexts.** With the rising popularity of pre-trained LLMs, it's natural to consider their role in physically grounded scenarios like robotics. Before the LLM surge, efforts were made to link vision and language (Das et al., 2017; Gordon et al., 2018; Shridhar et al., 2020; Duan et al., 2020).

Most of these were in embodied contexts, where agents answered questions by exploring environments (Das et al., 2017; Gordon et al., 2018), performed tasks with sub-steps (Shridhar et al., 2020), or rearranged scenes given instructions (Szot et al., 2022). Unlike these tasks which focus on visual and semantic attributes, NEWTON challenges language models with questions about explicit/implicit physical object properties, some beyond visual inference. More recently, LLMs are being explored for grounding robotic manipulation, often generating code or instructions for actions/skills (Liang et al., 2022; Ahn et al., 2022). Yet, there's a gap in using LLMs for object-centric physical reasoning in manipulation, which stems from limited evidence of LLMs' grounding abilities. Our work focuses on this by exploring and evaluating such abilities in physically grounded settings.

## 7 Conclusion

In this work, we present NEWTON, a **repository**, **pipeline**, and **benchmark** to support the assessment and refinement of LLMs in the context of physical attribute understanding of everyday objects. The NEWTON repository offers a large collection of object-attribute pairs, enabling the generation of infinite test scenarios for evaluating language models, using the NEWTON pipeline. To demonstrate its usefulness and potential, we introduce the NEWTON benchmark, which consists of 160K questions involving 700+ objects and 8 attributes. These questions cover three distinct tracks: fundamental object understanding, explicit application, and implicit analysis. NEWTON is built to enable comprehensive and automatic evaluation of language models across an array of scenarios, and support the reliable incorporation of LLMs into physically grounded contexts and applications.

## Limitations

While this paper sheds light on language models' physics reasoning abilities, it's crucial to note its limitations. Firstly, data collection through crowdsourcing introduces potential human errors, despite extensive filtering efforts by the authors. Secondly, the dataset's categories don't encompass the full range of real-world objects, limiting evaluation scenarios to those described. Thirdly, the paper doesn't explore different language models' suitability for various prompting strategies. Nonetheless, NEWTON could potentially be adapted for such evaluations in the realm of physical reasoning. Despite these limitations, we hope NEWTON encourages further research in benchmarking physical reasoning and enriching language models' understanding in a physically grounded context.

## Acknowledgements

The authors would like to thank members of the Personal Robotics Lab (PRL), Robotics and State Estimation Lab (RSELab), and UW NLP Group for fruitful discussions and insightful feedback on the manuscript. Yi Ru Wang is supported by the Natural Sciences and Engineering Research Council of Canada (NSERC). This work was (partially) funded by the National Science Foundation NRI (#2132848) and CHS (#2007011), DARPA RACER (#HR0011-21-C-0171), the Office of Naval Research (#N00014-17-1-2617-P00004, #2022-016-01 UW, and #1140209-405780), and Amazon.

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

# A Appendix

## A.1 NEWTON Benchmark Statistics

NEWTON has three main tracks: fundamental attribute understanding, explicit application, and implicit scenario-based analysis. The three tracks comprise of a total of 160K questions, covering 700+ objects and 8 attributes. Each object is annotated with 1-8 tags of object attributes. We present visualizations of the data-statistics for Track 1, 2, and 3 in Figure 5, 6, and 7, respectively. Figure 5 illustrates the data distribution of questions and attributes, total number of tokens per attribute, and number of attributes remaining after filtering for each object category. Figure 6 illustrates the data distribution for the explicit application track, and includes an analysis on the distribution of the top 100 highest occurring categories, percentage distribution of different attribute occurrences within the track, counts of question polarity and type, and total tokens for the questions in each attribute. Figure 7 illustrates the data distribution for the implicit scenario-based analysis track, and includes a percentage fraction of different scenarios, and the total number of tokens for questions in each scenario. Taking a closer look, we can see the initial set of attributes have a bias towards the elasticity, sharpness, softness, and malleability attributes, due to the process of filtering inconsistencies. To ensure a balanced dataset, template formation of Track 2 questions involved an additional re-sampling step to ensure the questions cover the attributes in a uniform way, hence why Track 2 questions are more uniform in nature. Track 3 scenarios are manually designed for each scenario, and hence data balance was not a primary focus.

## A.2 Detailed Results for NEWTON Benchmark Track 2

Track 2 involves a mixture of question types (Boolean, multiple choice), and questions which differ by polarity (positive, negative). We provide a detailed breakdown of results by polarity for models evaluated on the NEWTON Benchmark, Track 2, in Table 9.

## A.3 Qualitative Results

We present an example for model outputs for the Foundational Attribute Understanding (Track 1) in Figure 8. Note the common mistakes which language models make include (1) a tendency to choose the middle option, (2) hallucination, and (3) inaccurate understanding. In Figure 9, we examine the common failure modes of language models for a set of 10 questions from Track 2: Explicit Attribute Understanding. We find that for most models, the failure mode occurs due to the tendency to hallucinate. For instance, in Figure 9, for question 6, 7, and 8, Dolly fails to provide an answer which reflects the given options.

## A.4 Query Templates for Language Models

Different models require different variations of prompt input, and the unsuitable prompt could lead to an inaccurate assessment of the capability of language models. Hence, we adapt the question template for different families of models to adhere to the most effective prompting strategy, as shown in Figure 10.

## A.5 Synthesizing Additional Challenge Sets with NEWTON

We show a snapshot of the generated dataset using the procedure for synthesizing additional challenge sets in Figure 11. Note that all objects also have corresponding 3D models and 2D thumbnails, as shown in Figure 12 should the user wish to extend the dataset to a vision-language setting.

## A.6 Comparing Agreement and Accuracy Scores

In Table 10, we show the performance of various models when evaluated using the accuracy measure. In comparison to Table 4, which provides the quantitative results using the agreement measure, there is an overall increase in the absolute percentages, since the upper-threshold for the maximum attainable value has increased from human agreement percentage to 100%. However, we note that the conclusions mentioned in Section 5.2 remain the same.

## A.7 Dataset Examples

In this section, we provide a snapshot of the dataset, as shown in Figure 12. Each generated question is tagged with corresponding choices, ground truth response, question polarity, and relevant 2D RGB thumbnails and 3D object models. While NEWTON is designed for prompting and assessing language models, potential extensions could also leverage the paired RGB and 3D models to assess visually grounded models.

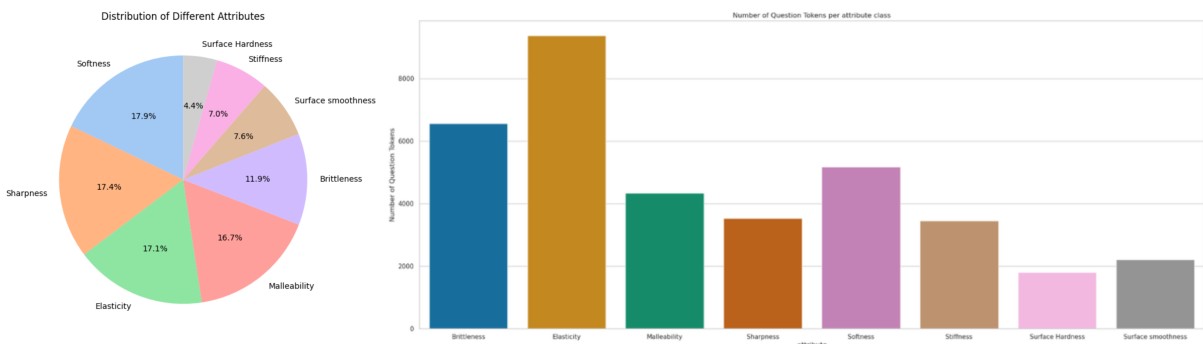

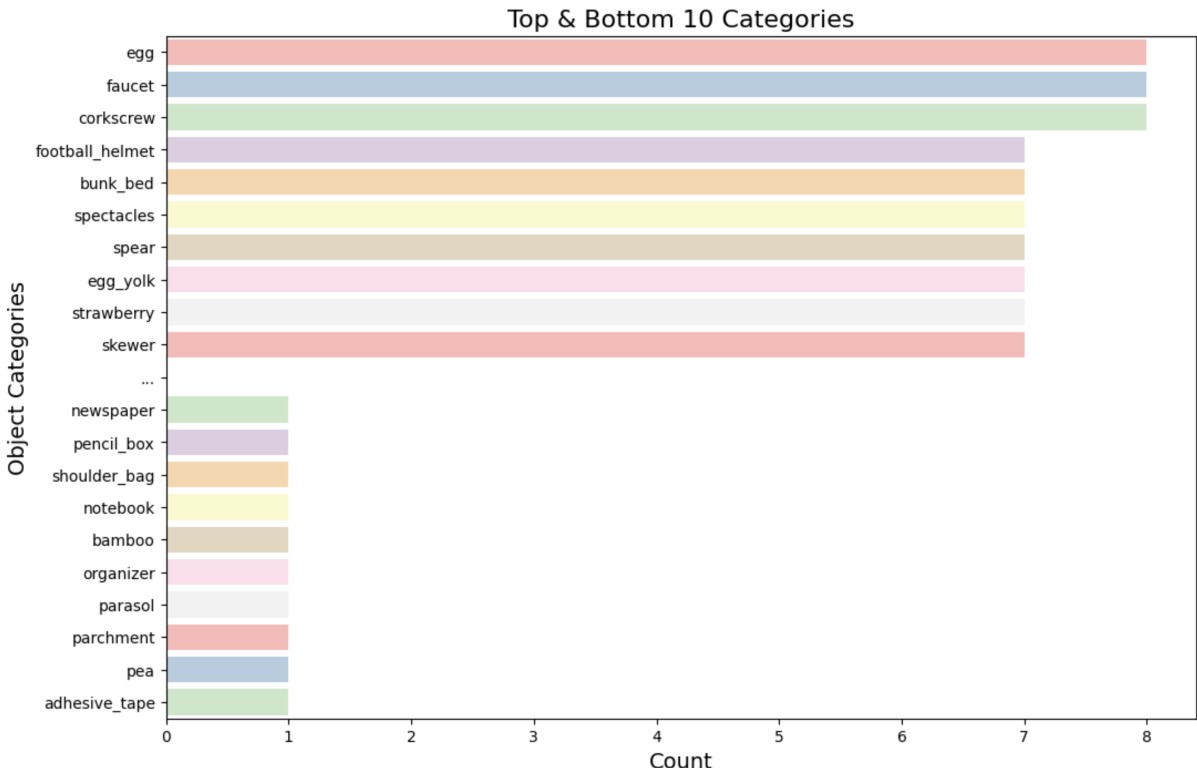

Figure 5. **NEWTON Benchmark Track 1** data statistics. We highlight the data distribution of questions and attributes, total number of tokens per attribute, and number of attributes remaining after filtering for each object category.

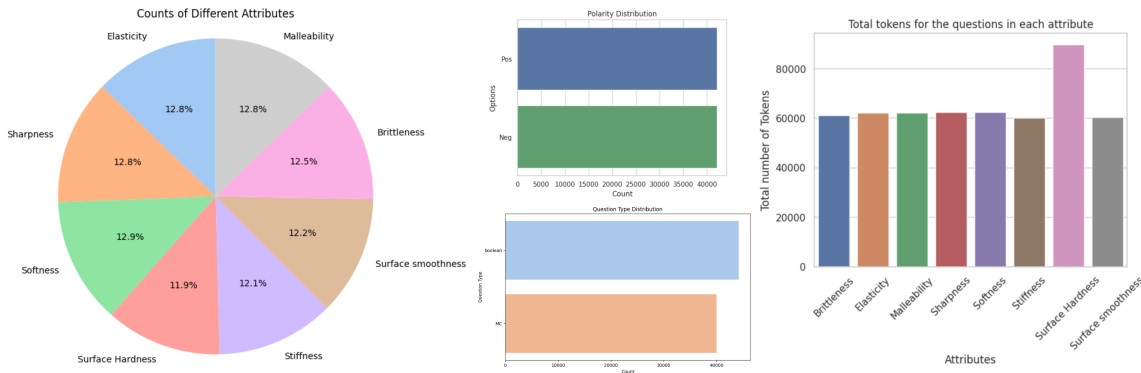

Figure 6. **NEWTON Benchmark Track 2** data statistics. We examine the data distribution for the explicit application track, and illustrate the percentage distribution of different attribute occurrences within the track, counts of question polarity and type, and total tokens for the questions in each attribute.

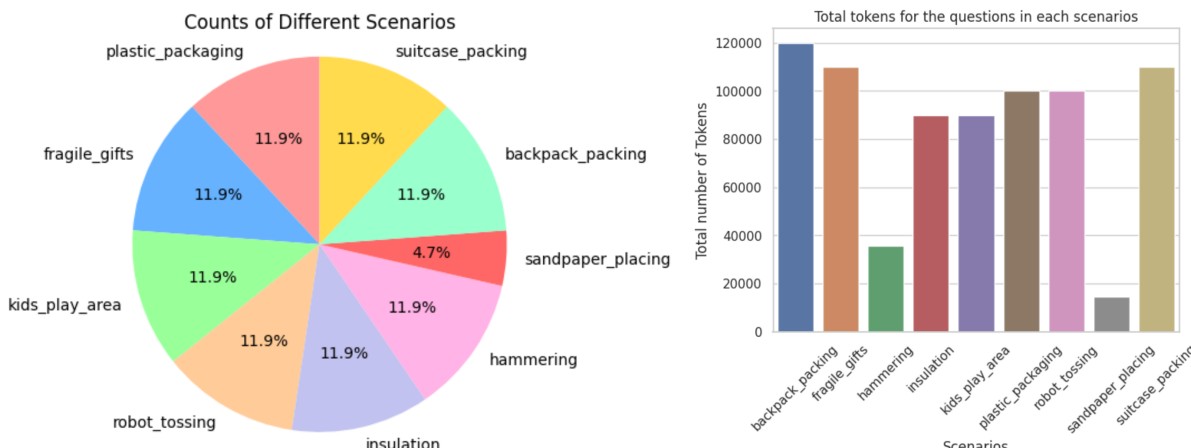

*Figure 7.* **NEWTON Benchmark Track 3** data statistics. Graphs show the percentage fraction of different scenarios, and the total number of tokens for questions in each scenario.

| Language Model | Elasticity | Stiffness | Surface Smoothness | Surface Hardness | Softness | Brittleness | Malleability | Sharpness | Overall |
|---|---|---|---|---|---|---|---|---|---|
| | | | | **Boolean (overall/positive/negative)** | | | | | |
| **Dolly-V2-7B** | 3.1 / 3.3 / 2.8 | 2.6 / 2.9 / 2.2 | 3.1 / 3.6 / 2.6 | 4.4 / 5.2 / 3.6 | 2.8 / 2.9 / 2.6 | 2.5 / 2.5 / 2.4 | 3.1 / 3.2 / 2.9 | 2.4 / 2.8 / 2.0 | 3.0 / 3.3 / 2.6 |
| **Flan-Alpaca-GPT-4-XL** | 54.8 / 57.7 / 52.0 | 61.5 / 73.4 / 49.6 | 49.8 / 51.6 / 48.0 | 59.5 / 67.8 / 51.1 | 58.1 / 54.3 / 62.0 | 49.2 / 43.9 / 54.4 | 48.8 / 45.9 / 51.6 | 68.1 / 77.0 / 59.2 | 56.2 / 58.8 / 53.7 |
| **Flan-T5-Small** | 53.5 / 53.1 / 53.9 | 50.3 / 52.2 / 48.4 | 48.5 / 52.0 / 45.0 | 49.3 / 51.7 / 46.9 | 51.4 / 48.4 / 54.4 | 48.3 / 43.6 / 53.0 | 52.4 / 48.1 / 56.7 | 55.4 / 52.3 / 58.6 | 51.2 / 50.1 / 52.3 |
| **Flan-T5-Base** | 36.8 / 43.0 / 30.6 | 35.3 / 37.8 / 32.7 | 34.8 / 37.6 / 31.9 | 34.6 / 30.0 / 39.3 | 30.8 / 32.2 / 29.4 | 40.8 / 32.2 / 49.4 | 37.1 / 34.4 / 39.8 | 26.3 / 30.2 / 22.4 | 34.5 / 34.7 / 34.3 |
| **Flan-T5-Large** | 42.3 / 42.3 / 42.4 | 40.1 / 40.3 / 40.0 | 43.8 / 43.9 / 43.7 | 20.6 / 16.3 / 24.9 | 50.2 / 53.3 / 47.0 | 29.8 / 30.2 / 29.3 | 38.8 / 39.5 / 38.2 | 52.1 / 57.3 / 46.9 | 40.1 / 40.8 / 39.3 |
| **Flan-T5-XL** | 61.4 / 66.0 / 56.8 | 60.4 / 54.1 / 66.8 | 50.5 / 50.9 / 50.1 | 65.3 / 57.4 / 73.2 | 74.0 / 78.1 / 69.9 | 51.4 / 51.7 / 51.2 | 59.2 / 67.8 / 50.6 | 78.8 / 79.2 / 78.4 | 62.9 / 63.6 / 62.1 |
| **UnifiedQA-V2-T5-Large** | 46.6 / 47.2 / 45.9 | 51.7 / 52.7 / 50.6 | 49.8 / 51.8 / 47.9 | 57.4 / 56.8 / 58.0 | 61.1 / 62.2 / 60.1 | 50.6 / 48.7 / 52.5 | 49.7 / 44.2 / 55.2 | 54.5 / 61.8 / 47.3 | 52.7 / 53.2 / 52.2 |
| **Alpaca-LoRa-7B** | 61.0 / 71.0 / 51.0 | 68.2 / 66.7 / 69.8 | 67.9 / 78.1 / 57.7 | 63.9 / 65.9 / 62.0 | 62.3 / 69.3 / 55.3 | 69.5 / 66.0 / 73.1 | 62.7 / 73.7 / 51.6 | 66.8 / 84.2 / 49.5 | 65.2 / 72.0 / 58.5 |
| **GPT-Turbo** | 49.5 / 41.3 / 57.7 | 58.5 / 62.5 / 54.4 | 51.6 / 53.2 / 50.1 | 54.4 / 56.5 / 52.2 | 67.5 / 67.2 / 67.8 | 51.7 / 45.9 / 57.4 | 61.5 / 66.7 / 56.3 | 81.5 / 81.8 / 81.2 | 59.8 / 59.6 / 60.0 |
| **GPT-4** | 66.6 / 63.0 / 70.1 | 61.7 / 70.1 / 53.4 | 60.8 / 61.6 / 60.0 | 66.8 / 74.4 / 59.1 | 78.0 / 74.8 / 81.1 | 66.8 / 64.5 / 69.0 | 71.5 / 70.2 / 72.7 | 82.3 / 81.4 / 83.2 | 69.6 / 70.1 / 69.1 |
| | | | | **Multiple Choice (overall/positive/negative)** | | | | | |
| **Dolly-V2-7B** | 14.4 / 13.7 / 15.2 | 9.8 / 10.4 / 9.1 | 11.7 / 12.0 / 11.4 | 13.2 / 10.6 / 15.8 | 12.6 / 12.7 / 12.4 | 11.7 / 12.5 / 10.8 | 11.4 / 9.4 / 13.4 | 10.6 / 9.6 / 11.6 | 11.9 / 11.4 / 12.5 |
| **Flan-Alpaca-GPT4-XL** | 76.6 / 83.2 / 70.0 | 79.6 / 77.6 / 81.6 | 37.2 / 27.5 / 46.9 | 80.5 / 80.2 / 80.7 | 78.3 / 84.9 / 71.7 | 22.6 / 30.8 / 14.4 | 48.8 / 45.9 / 51.6 | 91.8 / 98.6 / 85.0 | 65.9 / 68.2 / 63.5 |
| **Flan-T5-Small** | 23.6 / 27.3 / 20.0 | 24.8 / 29.5 / 20.0 | 25.1 / 24.3 / 26.0 | 23.9 / 26.3 / 21.6 | 23.7 / 24.8 / 22.6 | 24.0 / 22.6 / 25.4 | 23.9 / 24.8 / 23.0 | 24.4 / 34.6 / 14.2 | 24.2 / 26.8 / 21.6 |
| **Flan-T5-Base** | 39.9 / 66.2 / 13.6 | 36.9 / 50.6 / 23.3 | 23.2 / 22.6 / 23.9 | 42.0 / 48.6 / 35.5 | 40.4 / 67.4 / 13.4 | 14.8 / 12.6 / 17.1 | 32.1 / 39.2 / 25.0 | 57.1 / 87.2 / 27.0 | 35.8 / 49.3 / 22.3 |
| **Flan-T5-Large** | 61.0 / 74.3 / 47.7 | 56.9 / 49.8 / 64.0 | 23.0 / 16.2 / 29.7 | 72.4 / 69.2 / 75.5 | 64.6 / 69.2 / 60.0 | 32.8 / 48.8 / 16.7 | 38.3 / 32.0 / 44.6 | 90.4 / 94.6 / 86.2 | 54.9 / 56.8 / 53.1 |
| **Flan-T5-XL** | 70.5 / 81.2 / 59.9 | 77.9 / 76.9 / 79.0 | 42.5 / 39.2 / 45.8 | 82.5 / 85.3 / 79.6 | 77.6 / 82.8 / 72.4 | 30.0 / 37.2 / 22.8 | 54.7 / 57.4 / 51.9 | 92.9 / 98.2 / 87.6 | 66.1 / 69.8 / 62.4 |
| **UnifiedQA-V2-T5-Large** | 61.7 / 79.6 / 43.9 | 59.8 / 55.2 / 64.3 | 28.8 / 25.9 / 31.6 | 71.3 / 67.3 / 75.3 | 63.7 / 70.6 / 56.8 | 27.4 / 29.6 / 25.2 | 50.3 / 40.6 / 60.0 | 75.1 / 85.1 / 65.2 | 54.8 / 56.7 / 52.8 |
| **Alpaca-LoRa-7B** | 26.2 / 27.0 / 25.4 | 27.8 / 28.6 / 27.0 | 26.6 / 26.6 / 26.5 | 26.6 / 26.6 / 26.5 | 26.3 / 26.8 / 25.8 | 24.3 / 25.7 / 22.9 | 24.6 / 21.8 / 27.4 | 31.2 / 36.5 / 25.9 | 26.4 / 27.0 / 25.8 |
| **GPT-Turbo** | 49.0 / 50.4 / 47.6 | 69.1 / 73.8 / 64.4 | 42.8 / 33.3 / 52.3 | 81.1 / 81.2 / 81.1 | 73.6 / 74.9 / 72.3 | 44.7 / 52.0 / 37.4 | 43.3 / 27.7 / 59.0 | 92.7 / 97.5 / 87.8 | 62.0 / 61.3 / 62.7 |
| **GPT-4** | 75.4 / 67.5 / 83.4 | 73.8 / 76.3 / 71.3 | 64.6 / 59.4 / 69.8 | 84.8 / 88.8 / 80.7 | 91.7 / 91.8 / 91.6 | 67.8 / 75.1 / 60.4 | 72.8 / 68.8 / 76.9 | 98.5 / 99.0 / 98.0 | 78.7 / 78.3 / 79.0 |
| | | | | **Combined (overall/positive/negative)** | | | | | |
| **Dolly-V2-7B** | 8.3 / 8.1 / 8.5 | 6.1 / 6.6 / 5.6 | 7.3 / 7.7 / 6.9 | 8.8 / 7.9 / 9.7 | 7.3 / 7.4 / 7.1 | 6.9 / 7.3 / 6.4 | 6.9 / 6.1 / 7.8 | 6.2 / 5.9 / 6.4 | 7.2 / 7.1 / 7.3 |
| **Flan-Alpaca-GPT4-XL** | 64.9 / 69.5 / 60.3 | 70.4 / 75.4 / 65.3 | 43.7 / 39.8 / 47.5 | 69.9 / 74.0 / 65.9 | 67.4 / 68.4 / 66.5 | 36.5 / 37.7 / 35.3 | 54.2 / 53.8 / 54.6 | 79.1 / 87.0 / 71.1 | 60.8 / 63.3 / 58.3 |
| **Flan-T5-Small** | 39.6 / 41.1 / 38.2 | 37.8 / 41.1 / 34.5 | 37.1 / 38.5 / 35.7 | 36.7 / 39.1 / 34.3 | 38.6 / 37.5 / 39.7 | 36.7 / 33.6 / 39.9 | 39.2 / 37.3 / 41.0 | 41.1 / 44.1 / 38.1 | 38.4 / 39.0 / 37.7 |
| **Flan-T5-Base** | 38.2 / 53.8 / 22.7 | 36.1 / 44.1 / 28.1 | 29.1 / 30.2 / 28.0 | 38.3 / 39.2 / 37.4 | 35.3 / 48.5 / 22.0 | 28.4 / 22.9 / 34.0 | 34.8 / 36.6 / 32.9 | 40.6 / 56.6 / 24.5 | 35.1 / 41.6 / 28.6 |
| **Flan-T5-Large** | 51.0 / 57.1 / 44.8 | 48.4 / 45.0 / 51.8 | 33.6 / 30.4 / 36.9 | 46.4 / 42.6 / 50.1 | 56.8 / 60.6 / 53.0 | 31.2 / 39.1 / 23.3 | 38.6 / 36.0 / 41.1 | 69.8 / 74.6 / 65.1 | 47.1 / 48.4 / 45.8 |
| **Flan-T5-XL** | 65.6 / 73.0 / 58.2 | 69.0 / 65.3 / 72.8 | 46.6 / 45.2 / 48.0 | 73.9 / 71.3 / 76.4 | 75.7 / 80.3 / 71.0 | 41.2 / 44.8 / 37.7 | 57.1 / 63.0 / 51.2 | 85.3 / 88.0 / 82.6 | 64.4 / 66.5 / 62.2 |
| **UnifiedQA-V2-T5-Large** | 53.6 / 62.2 / 45.0 | 55.6 / 54.0 / 57.3 | 39.6 / 39.2 / 39.9 | 64.3 / 62.0 / 66.6 | 62.3 / 66.1 / 58.6 | 39.5 / 39.6 / 39.5 | 50.0 / 42.5 / 57.5 | 64.1 / 72.6 / 55.6 | 53.7 / 54.9 / 52.5 |
| **Alpaca-LoRa-7B** | 44.9 / 50.6 / 39.1 | 48.4 / 48.0 / 48.8 | 46.5 / 51.1 / 42.0 | 45.4 / 46.4 / 44.4 | 45.7 / 49.7 / 41.7 | 48.0 / 46.8 / 49.2 | 45.0 / 49.6 / 40.4 | 50.3 / 62.1 / 38.6 | 46.8 / 50.6 / 42.9 |
| **GPT-Turbo** | 49.3 / 45.5 / 53.1 | 63.7 / 68.0 / 59.3 | 47.3 / 43.5 / 51.2 | 67.7 / 68.8 / 66.5 | 70.3 / 70.7 / 69.9 | 48.4 / 48.8 / 47.9 | 53.0 / 48.6 / 57.5 | 86.7 / 89.1 / 84.2 | 60.9 / 60.4 / 61.3 |
| **GPT-4** | 70.7 / 65.1 / 76.2 | 67.6 / 73.1 / 62.2 | 62.7 / 60.5 / 64.8 | 75.7 / 81.6 / 69.9 | 84.3 / 82.6 / 86.0 | 67.2 / 69.6 / 64.9 | 72.1 / 69.5 / 74.7 | 89.8 / 89.6 / 90.1 | 73.9 / 74.0 / 73.8 |

*Table 9.* **Track 2: Explicit application** evaluation results on various language models. We separate the questions into two streams, Boolean, which consists of True/False style questions, and Multiple Choice, which consist of QA style questions with four answer choices. We report the model accuracy across each stream, as well as the combined accuracy, separated by the polarity of the question. For each stream, we report an averaged accuracy percentage across all models for each physical reasoning attribute. We also report an averaged accuracy percentage across all attributes, to gauge the overall understanding of language models across all physical reasoning attributes.

| Language Model | | | | Agreement (%) | | | | | |
|---|---|---|---|---|---|---|---|---|---|
| | Elasticity | Stiffness | Surface Smoothness | Surface Hardness | Softness | Brittleness | Malleability | Sharpness | Overall |
| **Dolly-V2-7B** | 26.0 | 15.3 | 5.5 | 4.7 | 1.4 | 1.4 | 10.0 | 2.6 | 8.6 |
| **Flan-Alpaca-GPT4-XL** | 11.2 | 1.5 | 75.5 | 25.8 | 28.4 | 7.5 | 12.0 | 8.2 | 18.3 |
| **Flan-T5-small** | 0.0 | 0.0 | 0.0 | 0.0 | 0.0 | 0.0 | 0.0 | 0.0 | 0.0 |
| **Flan-T5-Base** | 88.8 | 58.6 | 78.6 | 36.7 | 28.8 | 7.5 | 87.8 | 5.2 | 48.5 |
| **Flan-T5-Large** | 0.0 | 0.5 | 78.6 | 39.1 | 28.8 | 7.5 | 12.2 | 4.6 | 16.6 |
| **Flan-T5-XL** | 11.2 | 5.4 | 19.1 | 53.9 | 66.3 | 7.5 | 12.4 | 59.6 | 31.3 |
| **UnifiedQA-V2-T5-Large** | 0.0 | 0.0 | 78.6 | 39.1 | 11.4 | 7.5 | 12.0 | 26.0 | 17.2 |
| **Alpaca-LoRa-7B** | 11.2 | 0.0 | 56.4 | 0.8 | 16.4 | 4.6 | 12.0 | 2.0 | 12.1 |
| **GPT-Turbo** | 26.0 | 0.0 | 0.9 | 16.4 | 7.5 | 7.0 | 67.6 | 1.2 | 18.9 |
| **GPT-4** | 57.8 | 10.3 | 17.7 | 58.6 | 66.3 | 92.5 | 50.6 | 64.8 | 57.1 |

*Table 10.* **Accuracy Performance of Models on Foundational Attribute Comprehension.** We report the accuracy percentage, computed as a percentage of responses which agree with the majority voted human response. The upper threshold is 100%, which represents perfect overlap with the ground truth (majority voted human response).

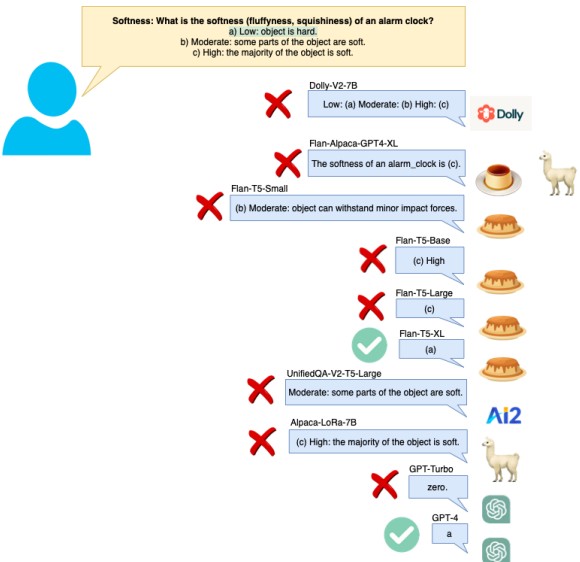

*Figure 8.* **Example of responses from NEWTON Track 1.** Note that the query is marked in the yellow text box, and the correct answer to choose is marked with a green highlight. Responses from different LLMs are shown in blue text-boxes, and correct responses are marked with a green check-mark.

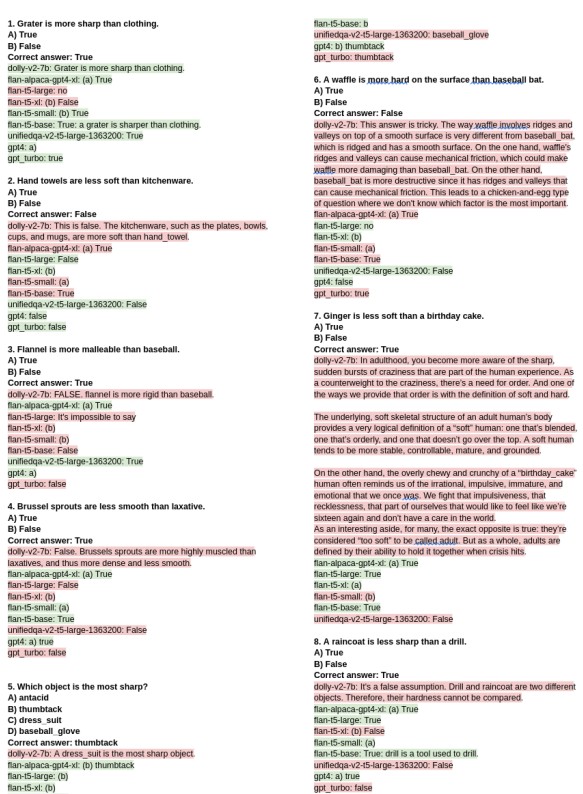

*Figure 9.* **Success vs. Failure Examples.** Typical failure modes of various models on a sample of NEWTON questions. Models typically fail due to hallucination and inaccurate understanding.

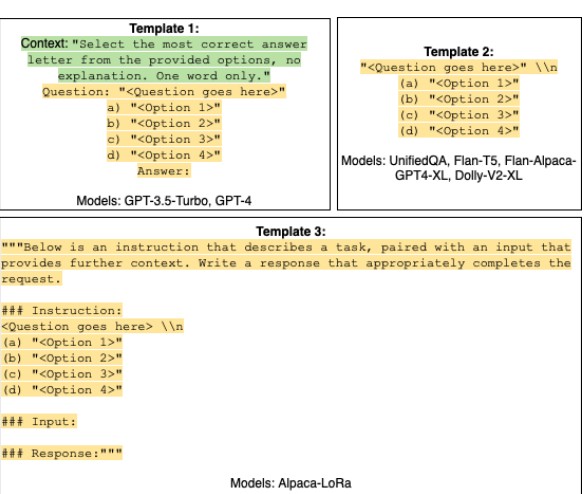

*Figure 10.* **Prompting Templates.** We show examples of the queries used for evaluating language models.

| Context | Question | Gt | Choice_1 | Choice_2 | Choice_3 | Choice_4 |
|---|---|---|---|---|---|---|
| When packing purchased objects into a box at a department store, | Which of amplifier, crescent_roll, crouton, flower_arrangement should be put at the bottom? | amplifier | amplifier | crescent_roll | crouton | flower_arrang |
| When packing purchased objects into a box at a department store, | Which of banner, ax, smoothie, oil_lamp should be put at the bottom? | ax | banner | ax | smoothie | oil_lamp |
| When packing purchased objects into a box at a department store, | Which of bagel, pudding, oil_lamp, barbell should be put at the bottom? | barbell | bagel | pudding | oil_lamp | barbell |
| When packing purchased objects into a box at a department store, | Which of bread, baseball_bat, candy, ring should be put at the bottom? | baseball_bat | bread | baseball_bat | candy | ring |
| When packing purchased objects into a box at a department store, | Which of basketball, cornmeal, bouquet, cornbread should be put at the bottom? | basketball | basketball | cornmeal | bouquet | cornbread |
| When packing purchased objects into a box at a department store, | Which of blueberry, whipped_cream, bath_mat, hamburger should be put at the bottom? | bath_mat | blueberry | whipped_crea | bath_mat | hamburger |
| When packing purchased objects into a box at a department store, | Which of omelet, cornmeal, baseball_cap, bell should be put at the bottom? | bell | omelet | cornmeal | baseball_cap | bell |
| When packing purchased objects into a box at a department store, | Which of blueberry, block, bubble_gum, banner should be put at the bottom? | block | blueberry | block | bubble_gum | banner |
| When packing purchased objects into a box at a department store, | Which of pizza, boot, pickle, egg_yolk should be put at the bottom? | boot | pizza | boot | pickle | egg_yolk |
| When packing purchased objects into a box at a department store, | Which of bow, sunflower, kiwi, hamburger should be put at the bottom? | bow | bow | sunflower | kiwi | hamburger |
| When packing purchased objects into a box at a department store, | Which of cigarette, nutcracker, breechcloth, bandanna should be put at the bottom? | breechcloth | cigarette | nutcracker | breechcloth | bandanna |
| When packing purchased objects into a box at a department store, | Which of envelope, green_bean, bucket, wind_chime should be put at the bottom? | bucket | envelope | green_bean | bucket | wind_chime |
| When packing purchased objects into a box at a department store, | Which of nutcracker, carnation, bunk_bed, lampshade should be put at the bottom? | bunk_bed | nutcracker | carnation | bunk_bed | lampshade |
| When packing purchased objects into a box at a department store, | Which of ballet_skirt, cabinet, bowler_hat, quiche should be put at the bottom? | cabinet | ballet_skirt | cabinet | bowler_hat | quiche |
| When packing purchased objects into a box at a department store, | Which of blackberry, ballet_skirt, cornmeal, cart should be put at the bottom? | cart | blackberry | ballet_skirt | cornmeal | cart |
| When packing purchased objects into a box at a department store, | Which of lampshade, crescent_roll, bandanna, checkerboard should be put at the bottom? | checkerboard | lampshade | crescent_roll | bandanna | checkerboard |
| When packing purchased objects into a box at a department store, | Which of garlic, chip, chocolate_mousse, blueberry should be put at the bottom? | chip | garlic | chip | chocolate_mo | blueberry |
| When packing purchased objects into a box at a department store, | Which of egg_roll, bowler_hat, chopping_board, bouquet should be put at the bottom? | chopping_boa | egg_roll | bowler_hat | chopping_boa | bouquet |
| When packing purchased objects into a box at a department store, | Which of sushi, snack_food, ring, cleaning_implement should be put at the bottom? | cleaning_impl | sushi | snack_food | ring | cleaning_impl |
| When packing purchased objects into a box at a department store, | Which of cleat, pie, meatball, wine_bottle should be put at the bottom? | cleat | cleat | pie | meatball | wine_bottle |
| When packing purchased objects into a box at a department store, | Which of plant, clothes_hamper, banana, ballet_skirt should be put at the bottom? | clothes_hamp | plant | clothes_hamp | banana | ballet_skirt |
| When packing purchased objects into a box at a department store, | Which of coatrack, tomato, envelope, cracker should be put at the bottom? | coatrack | coatrack | tomato | envelope | cracker |
| When packing purchased objects into a box at a department store, | Which of clementine, cornmeal, blackberry, coin should be put at the bottom? | coin | clementine | cornmeal | blackberry | coin |
| When packing purchased objects into a box at a department store, | Which of combination_lock, peach, laxative, pancake should be put at the bottom? | combination_ | combination_ | peach | laxative | pancake |
| When packing purchased objects into a box at a department store, | Which of peach, crouton, sawhorse, laxative should be put at the bottom? | sawhorse | peach | crouton | sawhorse | laxative |
| When packing purchased objects into a box at a department store, | Which of pastry, scarf, green_onion, plant should be put at the bottom? | scarf | pastry | scarf | green_onion | plant |
| When packing purchased objects into a box at a department store, | Which of green_bean, kite, scissors, sunflower should be put at the bottom? | scissors | green_bean | kite | scissors | sunflower |
| When packing purchased objects into a box at a department store, | Which of scraper, carnation, envelope, kiwi should be put at the bottom? | scraper | scraper | carnation | envelope | kiwi |
| When packing purchased objects into a box at a department store, | Which of ice_cream, blueberry, screwdriver, snack_food should be put at the bottom? | screwdriver | ice_cream | blueberry | screwdriver | snack_food |
| When packing purchased objects into a box at a department store, | Which of green_bean, shawl, pickle, chocolate_cake should be put at the bottom? | shawl | green_bean | shawl | pickle | chocolate_cak |
| When packing purchased objects into a box at a department store, | Which of wind_chime, persimmon, cupcake, shears should be put at the bottom? | shears | wind_chime | persimmon | cupcake | shears |

*Figure 11.* **Snapshot of generated challenge-set example.** In this example, we demonstrate synthesis of a multiple choice challenge set. We see that each synthesized sample has context, question, and ground truth annotations.

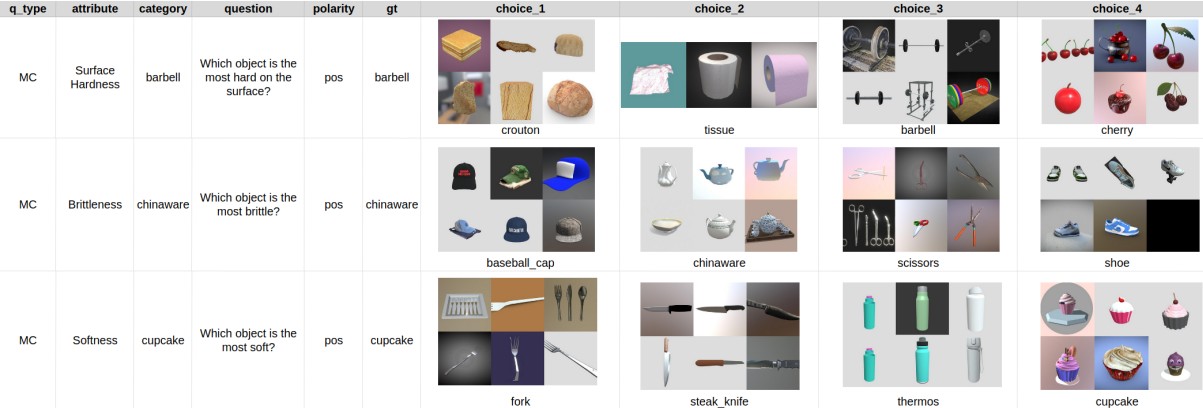

| q_type | attribute | category | question | polarity | gt | choice_1 | choice_2 | choice_3 | choice_4 |
|---|---|---|---|---|---|---|---|---|---|
| MC | Surface Hardness | barbell | Which object is the most hard on the surface? | pos | barbell | crouton | tissue | barbell | cherry |
| MC | Brittleness | chinaware | Which object is the most brittle? | pos | chinaware | baseball_cap | chinaware | scissors | shoe |
| MC | Softness | cupcake | Which object is the most soft? | pos | cupcake | fork | steak_knife | thermos | cupcake |

*Figure 12.* **Dataset Sample.** We provide a snapshot of the NEWTON Benchmark. Questions are populated automatically with meaningful object types. Each question is labelled with the attribute, object category, ground-truth object, question polarity, question type, and several candidate object options. Each candidate object also has several corresponding 3D object models, should the user wish to use NEWTON with perception in the loop.