# OpenReview forum: "NEWTON: Are Large Language Models Capable of Physical Reasoning?"
_EMNLP/2023/Conference — EMNLP 2023 Findings_

### Official Review · Reviewer_G7Xo · 2023-08-04

**Typos Grammar Style And Presentation Improvements:** No
**Soundness:** 3

**Excitement:**

3: Ambivalent: It has merits (e.g., it reports state-of-the-art results, the idea is nice), but there are key weaknesses (e.g., it describes incremental work), and it can significantly benefit from another round of revision. However, I won't object to accepting it if my co-reviewers champion it.

**Missing References:**

EQA paper as mentioned above.

**Paper Topic And Main Contributions:**

The paper proposes a new benchmark, called Newton, that systematically studies the problem of physical reasoning, which involves the cognitive process of comprehending and predicting the actions and dynamics of physical systems based on observation and common sense. The existing benchmark does not study the problem of physical reasoning. The dataset consists of 700 objects across 8 physical attributes and 160K pre-generated questions. In addition, the dataset creation process consists of 4 main components: (1) annotation that let annotators to best describe the property of the object; (2) filtering that eliminates ambiguous data points; (3) template generation that generates the question, and (4) evaluation that benchmarks existing LLMs.  In addition, the questions of interest consist of (1) Foundational Attribute Understanding that concerns the physical property of the objects, (2) Explicit Application that groups the object class, and (3) Implicit Scenario-based Analysis that chooses the object based on the condition.

Finally, several conclusions are made. First, LLMs do not have a good comprehension of the physical concept of objects and are capable of decision-making in implicit scenario-based tasks. And few ablation studies are also provided such as fine-tuning the model can improve the performance.

**Questions For The Authors:**

1. Could you please try some experiments to see if the prompting approach can improve the performance?

2. I want to learn more about what is the difference between this paper and existing EQA tasks.

3. The paper makes a claim that the dataset also studies how observations affected decision marking. Does this dataset contain the visual image shown in Figure 1? Or it is purely the text?

**Reasons To Accept:**

1. The paper is well-written. It has a clear motivation and the experiment section

2. The experiments make sense to me, and ablation studies make the argument stronger. For example, I appreciate that the author fine-tune their models on this dataset.

3. This paper provides a systematic way to study the problem of physical reasoning.

**Reasons To Reject:**

1. Lack of awareness of work called "Embodied Question Answering (EQA)". The major issue of this paper is that it does not discuss the relationship between the concept of physical reasoning and EQA tasks (https://openaccess.thecvf.com/content_cvpr_2018/papers/Das_Embodied_Question_Answering_CVPR_2018_paper.pdf). In EQA, the robotic agent navigates rich 3D environments to answer questions such as "what is the color of the ball?" or "how many apples in the kitchen", which needs language understanding, visual reasoning, and goal-driven navigation to succeed. For Track 3: Implicit Scenario-based Analysis questions such as I am packing a backpack. Question: Which of <object 1>, <object 2>, <object 3>, <object 4> should I put at the bottom?, this task feels very similar to the nav-pick-place task in EQA (see this paper: https://aihabitat.org/challenge/rearrange_2022/). I think the author should add more discussions about this.

2. Need more experience to verify the argument "current LLMs do not have physical reasoning capability". For example, I am curious to see if we provide a prompt, i.e., a few examples, will the model have better performance? I think for the current capability of GPT4, the number will be higher than the results reported in the paper.

3. The scope of physical reasoning. I could imagine that there might be some physical property does not catch by this dataset. Could the instructions provided in the paper consider all the physical properties?

**Reproducibility:**

4: Could mostly reproduce the results, but there may be some variation because of sample variance or minor variations in their interpretation of the protocol or method.

**Reviewer Confidence:**

3: Pretty sure, but there's a chance I missed something. Although I have a good feel for this area in general, I did not carefully check the paper's details, e.g., the math, experimental design, or novelty.

---

> ### Author Rebuttal · Authors · 2023-08-28
>
> **We thank the reviewer for their insightful comments and questions, and provide our responses below:**
>
> > *"Lack of awareness of work called "Embodied Question Answering (EQA)"*
>
> Thanks to the reviewer for pointing this out! We have modified the related works, where we discuss the relevance and differences between embodied question answering and our work. In particular, embodied question answering is centered around questions with answers visually grounded to a physical environment. While certain physical attributes can be differentiated explicitly, there are implicit attributes which cannot be determined purely through visual cues (i.e. elasticity, stiffness, etc.), and requires some degree of common-sense reasoning. Our work is focused specifically on physical attributes on an object level, by curating a large repository of object-attribute pairs. This can potentially complement existing EQA benchmarks, by augmenting questions with an additional layer of physical reasoning.
>
> > *"Need more experience to verify the argument "current LLMs do not have physical reasoning capability". For example, I am curious to see if we provide a prompt, i.e., a few examples, will the model have better performance? I think for the current capability of GPT4, the number will be higher than the results reported in the paper."* and *"Could you please try some experiments to see if the prompting approach can improve the performance?"*
>
> We thank the reviewer for the suggestion of experimenting with prompt engineering! We have conducted additional experiments and highlighted the results in the following tables (Boolean, Multiple Choice, Overall)
>
> **Boolean:**
>
> | **LLM + Prompting Method** | Elasticity | Stiffness | Surface Smoothness | Surface Hardness | Softness | Brittleness | Malleability | Sharpness | Overall |
> | ------------------------ | ---------- | --------- | ----------------- | --------------- | -------- | ----------- | ----------- | --------- | ------- |
> | **GPT-Turbo + V1**        | 51.5       | 60.5      | 53.0             | 56.0            | 71.0     | 53.5        | 57.0        | 80.5      | 60.4    |
> | **GPT-Turbo + V2**        | 56.5       | 71.5      | 64.0             | 52.0            | 49.5     | 68.5        | 56.5        | 61.0      | 59.9    |
> | **GPT-Turbo + V3**        | 54.0       | 53.5      | 59.0             | 60.5            | 59.0     | 61.0        | 56.0        | 60.0      | 57.9    |
> | **GPT-Turbo + Ensemble**  | 55.0       | 68.5      | 64.5             | 60.5            | 58.0     | 62.5        | 57.0        | 69.0      | 61.9    |
> | **GPT-Turbo + Instruction**| 62.0       | 68.0      | 56.5             | 68.0            | 57.5     | 55.0        | 66.0        | 80.5      | 64.2    |
> | **GPT-4 + V1**            | 68.5       | 67.0      | 66.5             | 65.5            | 82.5     | 63.5        | 73.0        | 86.0      | 71.6    |
> | **GPT-4 + V2**            | 66.0       | 80.0      | 58.0             | 72.0            | 71.5     | 83.5        | 71.5        | 69.5      | 71.5    |
> | **GPT-4 + V3**            | 75.5       | 71.5      | 61.0             | 88.5            | 73.0     | 88.0        | 72.0        | 83.5      | 76.6    |
> | **GPT-4 + Ensemble**      | 68.5       | 76.5      | 61.5             | 79.0            | 77.0     | 85.5        | 75.0        | 83.0      | 75.8    |
> | **GPT-4 + Instruction**   | 77.5       | 85.5      | 75.5             | 90.5            | 80.0     | 68.5        | 72.0        | 96.5      | 80.8    |
>
> **Multiple choice:**
>
> | **LLM + Prompting Method** | Elasticity | Stiffness | Surface Smoothness | Surface Hardness | Softness | Brittleness | Malleability | Sharpness | Overall |
> | ------------------------ | ---------- | --------- | ----------------- | --------------- | -------- | ----------- | ----------- | --------- | ------- |
> | **GPT-Turbo + V1**        | 46.5       | 72.5      | 48.5             | 82.5            | 78.0     | 41.5        | 41.0        | 92.0      | 62.8    |
> | **GPT-Turbo + V2**        | 25.5       | 56.0      | 36.5             | 52.5            | 55.0     | 44.5        | 38.5        | 61.5      | 46.2    |
> | **GPT-Turbo + V3**        | 54.5       | 55.0      | 31.5             | 69.5            | 53.5     | 57.0        | 36.0        | 66.5      | 52.9    |
> | **GPT-Turbo + Ensemble**  | 36.5       | 54.5      | 33.0             | 67.0            | 56.5     | 45.0        | 34.0        | 65.0      | 48.9    |
> | **GPT-Turbo + Instruction**| 30.5       | 53.5      | 34.0             | 63.5            | 49.5     | 32.0        | 25.0        | 69.5      | 44.7    |
> | **GPT-4 + V1**            | 74.0       | 76.5      | 70.0             | 87.5            | 91.5     | 65.0        | 71.0        | 98.0      | 79.2    |
> | **GPT-4 + V2**            | 49.5       | 66.5      | 37.5             | 83.5            | 70.0     | 73.5        | 69.0        | 82.0      | 66.4    |
> | **GPT-4 + V3**            | 75.5       | 68.0      | 40.5             | 84.0            | 75.0     | 78.5        | 70.5        | 76.0      | 71.0    |
> | **GPT-4 + Ensemble**      | 65.5       | 68.5      | 47.0             | 84.5            | 81.5     | 77.0        | 71.5        | 87.5      | 72.9    |
> | **GPT-4 + Instruction**   | 62.5       | 57.5      | 54.0             | 71.5            | 65.0     | 50.5        | 31.0        | 74.0      | 58.2    |
>
>
> **Overall:**
> | **LLM + Prompting Method** | Elasticity | Stiffness | Surface Smoothness | Surface Hardness | Softness | Brittleness | Malleability | Sharpness | Overall |
> | ------------------------ | ---------- | --------- | ----------------- | --------------- | -------- | ----------- | ----------- | --------- | ------- |
> | **GPT-Turbo + V1**        | 49.0       | 66.5      | 50.7             | 69.2            | 74.5     | 47.5        | 49.0        | 86.2      | 61.6    |
> | **GPT-Turbo + V2**        | 41.0       | 63.7      | 50.2             | 52.2            | 52.2     | 56.5        | 47.5        | 61.3      | 53.1    |
> | **GPT-Turbo + V3**        | 54.2       | 54.2      | 45.2             | 65.0            | 56.2     | 59.0        | 46.0        | 63.2      | 55.4    |
> | **GPT-Turbo + Ensemble**  | 45.8       | 61.5      | 48.8             | 63.7            | 57.2     | 53.8        | 45.5        | 67.0      | 55.4    |
> | **GPT-Turbo + Instruction**| 46.2      | 60.8      | 45.2             | 65.8            | 53.5     | 43.5        | 45.5        | 75.0      | 54.4    |
> | **GPT-4 + V1**            | 71.2       | 71.8      | 68.2             | 76.5            | 87.0     | 64.2        | 72.0        | 92.0      | 75.4    |
> | **GPT-4 + V2**            | 57.8       | 73.2      | 47.8             | 77.8            | 70.8     | 78.5        | 70.2        | 75.8      | 69.0    |
> | **GPT-4 + V3**            | 75.5       | 69.8      | 50.7             | 86.2            | 74.0     | 83.2        | 71.2        | 79.8      | 73.8    |
> | **GPT-4 + Ensemble**      | 67.0       | 72.5      | 54.2             | 81.8            | 79.2     | 81.2        | 73.2        | 85.2      | 74.3    |
> | **GPT-4 + Instruction**   | 70.0       | 71.5      | 64.8             | 81.0            | 72.5     | 59.5        | 51.5        | 85.2      | 69.5    |
>
> In total, we experiment with 5 methods of prompting, including three paraphrased variations of the initial prompt, prompt ensembling using the three variations (finding and using the most frequent occurring response), and example-based prompting (providing 8 unique examples in the prompt). We find that for boolean style questions, ensemble and example-based prompting leads to improved performance for GPT-Turbo and GPT-4 for most physical attributes. However, for multiple choice questions, such methods yield similar or worse performance as compared to directly prompting. This outcome demonstrates the importance of prompt engineering, and its varying impact on different types of tasks. An interesting future direction is to learn and find the most suitable type of prompt given different question styles and attribute topics to maximize the model’s accuracy.
>
> > *"The scope of physical reasoning. I could imagine that there might be some physical property does not catch by this dataset. Could the instructions provided in the paper consider all the physical properties?"*
>
> We thank the reviewer for bringing up this important point! While we tried to be as comprehensive as possible in selecting the physics attributes considered, there indeed exists many other variations of attributes which are of importance to different applications. It is essential to recognize that the versatility embedded within NEWTON’s design facilitates seamless integration of further attributes. For instance, new attributes could be incorporated into the dataset through the same procedure described in Figure 2, involving annotation and filtering. Given our intention to make the annotation interface openly accessible, the acquisition of additional annotations can be done in a streamlined manner. This approach not only facilitates augmentation of the dataset with new attributes, but also promotes a convenient and collaborative means of enhancing NEWTON’s comprehensiveness.
>
> > *"I want to learn more about what is the difference between this paper and existing EQA tasks."*
>
> We thank the reviewer for this question. Existing embodied question answering tasks primarily take a perception-oriented approach in understanding objects and scenes. Tasks typically include searching for items near another item, determining the visual characteristics of a particular object, among others. Our benchmark focuses on understanding the physical attributes of objects, some of which are implicit (i.e. elasticity, brittleness, etc.) and cannot be determined through typical perception means (RGB, RGB-D, etc.). NEWTON can potentially complement existing EQA tasks by adding an additional layer of QA tasks involving physics reasoning in scenes, and can help with informed incorporation of language models in physically grounded settings (i.e. robot manipulation tasks).
>
> > *"The paper makes a claim that the dataset also studies how observations affected decision marking. Does this dataset contain the visual image shown in Figure 1? Or it is purely the text?"*
>
> We thank the reviewer for bringing up this point! The proposed dataset not only has language-based annotations, but also has annotations coupled with the object category, thumbnails (2D image), and 3D models. While incorporating and benchmarking with the provided visual information is beyond the scope of the current paper, our dataset is capable of supporting evaluation of Vision Language models, in addition to purely language models. We have updated the paper with examples of the questions, associated object categories, and paired images in Appendix A.9 and Figure 12, all of which will be made available upon notification of results.

---

### Official Review · Reviewer_EC6n · 2023-08-05

**Soundness:** 2

**Excitement:**

3: Ambivalent: It has merits (e.g., it reports state-of-the-art results, the idea is nice), but there are key weaknesses (e.g., it describes incremental work), and it can significantly benefit from another round of revision. However, I won't object to accepting it if my co-reviewers champion it.

**Paper Topic And Main Contributions:**

The paper proposes Newton, a new benchmark to evaluate model's physical reasoning capabilities. The data collection is performed through a crowdsourcing process using the Label Studio Enterprise platform. Each annotation task consists of a question, a set of images, and three answer choices. A minimum of four annotators contribute overlapping annotations for each object-attribute pair, and a stringent filtering threshold of 0.75 is applied to ensure reliability. Over 20,000 annotations were collected, featuring 700+ objects and 8 unique physical reasoning attributes. After filtering, there are 160K questions in the benchmark. The authors evaluate 10 recent language models on the benchmark. They find that LLMs overall do not have a consistent comprehension of foundamental physical attribute, while some model (e.g. GPT4) can perform quite well on implicit scenario-based analysis where the reasoning of physical attribute is used to make decision under certain situation.

**Questions For The Authors:**

A: Is there a particular reason why the Agreement metric is used to evaluate Track 1 while the Accuracy metric is used to evaluate Track 2 and 3?

B: Are the polarity and position bias problem addressed in the benchmark evaluation results (i.e. scores in table 4 & 5), or only be taken care in the ablation study?

**Reasons To Accept:**

- The idea of setting up evaluation benchmarks for physical reasoning capability of LLMs is well-motivated. The proposed resource is a magnitude larger in terms of the number of questions than existing benchmarks, and also covers more objects and physical attributes.

**Reasons To Reject:**

- The question can be ambiguous without giving a specific instance of the object, as the object can be made of different materials that leads to different physical properties. For example,  a metal bowl and a glass bowl may have different degree of brittleness, while a flat and a inflated basketball may have different degree of sftiffness. As a result, asking the physical property for a general object type does not make sense.
- The metrics (as described in Appendix A.1) are hard to understand. As a result, it is not clear what the numbers mean in the result tables. For example, if the accuracy for binary boolean questions in Table 5 is measured by the number of correct answers divided by the total number of questions, simply reversing the prediction of Dolly-V2-7B would yield an accuracy of 97\% which is clearly wrong.

**Reproducibility:**

2: Would be hard pressed to reproduce the results. The contribution depends on data that are simply not available outside the author's institution or consortium; not enough details are provided.

**Reviewer Confidence:**

3: Pretty sure, but there's a chance I missed something. Although I have a good feel for this area in general, I did not carefully check the paper's details, e.g., the math, experimental design, or novelty.

**Typos Grammar Style And Presentation Improvements:**

- The captions are too small. Text in Figure 3 is hard to see. The data statistics in figure 4 and 6 are hard to see.

---

> ### Author Rebuttal · Authors · 2023-08-28
>
> We thank the reviewer for their helpful suggestions and feedback, and address all the concerns below:
>
> > *"The question can be ambiguous without giving a specific instance of the object, as the object can be made of different materials that leads to different physical properties. For example, a metal bowl and a glass bowl may have different degree of brittleness, while a flat and a inflated basketball may have different degree of sftiffness. As a result, asking the physical property for a general object type does not make sense."*
>
> We thank the reviewer for bringing up this point, and agree that for some objects, different objects made of different materials will exhibit different physical properties, and hence may raise ambiguities. We note that this was also a main consideration when we implemented the pipeline, and hence motivated some of the design choices for the crowdsourcing and data filtering stages. During the annotation crowdsourcing stage, we designed the interface such that annotators are provided with a set of images which represent relevant objects for the specific object type (as seen in Figure 2). This enabled the annotations to be grounded on common objects within the category. To further disambiguate the responses, we enforce a minimum of 0.75 inter-annotator agreement for all annotated object-attribute pairs. For particular attributes/objects which exhibit high variance in responses across annotators, we exclude them from the subsequent phases of the benchmark QA dataset synthesis. Understanding that the nature of physical reasoning holds inherent subjectivity, our intention is to mitigate this subjectivity through our methodology. By grounding human annotations on objects (which inherently captures possible state and material variations), and by enforcing high inter-annotator agreement, we hope to reduce the presence of subjectivity within the collected annotations. Furthermore, in our pursuit of minimizing subjectivity in annotating physical attributes of objects, a prospective avenue involves breaking down objects into constituent parts and annotating at a part-level granularity. This is an ongoing next step.
>
> > *"The metrics (as described in Appendix A.1) are hard to understand. As a result, it is not clear what the numbers mean in the result tables. For example, if the accuracy for binary boolean questions in Table 5 is measured by the number of correct answers divided by the total number of questions, simply reversing the prediction of Dolly-V2-7B would yield an accuracy of 97% which is clearly wrong."*
>
> Thanks for raising this question! We have modified the paper with a more in-depth explanation of the metrics in the Appendix A.1, this updated version will be made available upon notification of results, per EMNLP Policy. The agreement metric regards the human agreement percentage as an upper limit and adjusts the calculated accuracy based on this percentage. This adjustment results in a higher emphasis on questions with substantial human agreement, while assigning comparatively less significance to questions where human annotator responses exhibit greater diversity. This metric aids in evaluating the extent to which language model responses align with those of humans. On the other hand, the accuracy metric considers the human-majority response as the definitive label and thus has a maximum attainable value of 100%. This metric gauges the proportion of responses that align with the majority-voted human response.
>
> To answer the question regarding Dolly, Dolly-V2-7B is an instruction-following language model. Similar to GPT-3.5-Turbo and GPT-4, the model’s output is generated in a free-form manner. This means that it may be possible to generate responses that are hallucinations, etc. Hence, the accuracy metric filters for answers which closely resemble the correct answer. Incorrect responses as an example could be: more than one answer given, invalid answer (answering c when only a and b are possible), or no response (i.e. N/A). Hence, we cannot reverse the predictions of the models, since it is often the case that the model does not output responses that can be reversed. We provide some examples of incorrect responses below:
>
> `A raincoat is less sharp than a drill.
> A) True
> B) False
> Correct answer: True`
>
> * **dolly-v2-7b**: It's a false assumption. Drill and raincoat are two different objects. Therefore, their hardness cannot be compared.
>
> `Ginger is less soft than a birthday cake.
> A) True
> B) False
> Correct answer: True`
>
> * **dolly-v2-7b**: In adulthood, you become more aware of the sharp, sudden bursts of craziness that are part of the human experience. As a counterweight to the craziness, there’s a need for order. And one of the ways we provide that order is with the definition of soft and hard. The underlying, soft skeletal structure of an adult human’s body provides a very logical definition of a “soft” human: one that’s blended, one that’s orderly, and one that doesn’t go over the top. A soft human tends to be more stable, controllable, mature, and grounded. On the other hand, the overly chewy and crunchy of a “birthday_cake” human often reminds us of the irrational, impulsive, immature, and emotional that we once was. We fight that impulsiveness, that recklessness, that part of ourselves that would like to feel like we’re sixteen again and don’t have a care in the world. As an interesting aside, for many, the exact opposite is true: they’re considered “too soft” to be called adult. But as a whole, adults are defined by their ability to hold it together when crisis hits.
>
> `A waffle is more hard on the surface than baseball bat.
> A) True
> B) False
> Correct answer: False`
>
> * **dolly-v2-7b:** This answer is tricky. The way waffle involves ridges and valleys on top of a smooth surface is very different from baseball_bat, which is ridged and has a smooth surface. On the one hand, waffle's ridges and valleys can cause mechanical friction, which could make waffle more damaging than baseball_bat. On the other hand, baseball_bat is more destructive since it has ridges and valleys that can cause mechanical friction. This leads to a chicken-and-egg type of question where we don't know which factor is the most important.
>
> As can be seen, Dolly occasionally outputs responses which are hallucinated and does not match any of the given boolean options. In these scenarios, it is not possible to simply reverse the prediction. We have provided additional examples of correct vs. incorrect responses in the updated Appendix A.4 and Figure 11, this will be made available in the paper upon notification of results.
>
>
> > *"Is there a particular reason why the Agreement metric is used to evaluate Track 1 while the Accuracy metric is used to evaluate Track 2 and 3?"*
>
> Thanks for bringing up this question. The reason that the Agreement metric is used to evaluate Track 1, while the Accuracy metric is used to evaluate Track 2 and 3 is due to the nature of the tracks. Track 1 is designed to assess whether the reasoning capabilities of language models resembles that of humans. Hence, for this track, we treat the human agreement as an upper bound, and scale the accuracy measures by the human agreement measure for each assessed question. Doing so, we are effectively putting more weight on the questions for which there was high human agreement, and less weight on questions where there were more variance across human annotators. Since the reviewer brought up this question, we introduce the following table, which measures the performance of various models on the Track 1 questions with the accuracy measure.
>
> | **Language Model** | **Agreement (%)** | | | | | | | | | | |
> | ------------------ | ----------------- | - | - | - | - | - | - | - | - | - | - |
> |                    | **Elasticity**    | **Stiffness** | **Surface Smoothness** | | **Surface Hardness** | **Softness** | **Brittleness** | | **Malleability** | **Sharpness** | **Overall** |
> | **Dolly-V2-7B**                    | 26.0  | 15.3  | 5.5  | | 4.7  | 1.4  | 1.4  | | 10.0  | 2.6  | 8.6  |
> | **Flan-Alpaca-GPT4-XL**             | 11.2  | 1.5   | 75.5 | | 25.8 | 28.4 | 7.5  | | 12.0  | 8.2  | 18.3 |
> | **Flan-T5-small**                  | 0.0   | 0.0   | 0.0  | | 0.0  | 0.0  | 0.0  | | 0.0   | 0.0  | 0.0  |
> | **Flan-T5-Base**                   | 88.8  | 58.6  | 78.6 | | 36.7 | 28.8 | 7.5  | | 87.8  | 5.2  | 48.5 |
> | **Flan-T5-Large**                  | 0.0   | 0.5   | 78.6 | | 39.1 | 28.8 | 7.5  | | 12.2  | 4.6  | 16.6 |
> | **Flan-T5-XL**                     | 11.2  | 5.4   | 19.1 | | 53.9 | 66.3 | 7.5  | | 12.4  | 59.6 | 31.3 |
> | **UnifiedQA-V2-T5-Large**          | 0.0   | 0.0   | 78.6 | | 39.1 | 11.4 | 7.5  | | 12.0  | 26.0 | 17.2 |
> | **Alpaca-LoRa-7B**                 | 11.2  | 0.0   | 56.4 | | 0.8  | 16.4 | 4.6  | | 12.0  | 2.0  | 12.1 |
> | **GPT-Turbo**                      | 26.0  | 0.0   | 0.9  | | 16.4 | 7.5  | 7.0  | | 67.6  | 1.2  | 18.9 |
> | **GPT-4**                          | 57.8  | 10.3  | 17.7 | | 58.6 | 66.3 | 92.5 | | 50.6  | 64.8 | 57.1 |
>
> We find that since the upper bound of performance is now 100% (as opposed to the human agreement percentage with the agreement metric), there is an overall increase with all metrics across all models. However, the general observed trend remains the same as Table 4, which uses the agreement metric, and the conclusions drawn in Section 5.2 remain.
>
> > *"Are the polarity and position bias problem addressed in the benchmark evaluation results (i.e. scores in table 4 & 5), or only be taken care in the ablation study?"*
>
> We thank the reviewer for raising this point. The polarity and position bias problem is specifically examined in the ablation study, where we recognized that the problem is common across the majority of examined language models. To delve into the influence of polarity in a more comprehensive manner, we provide Table 8 in the Appendix, which scrutinizes the models’ performance in scenarios where the polarity is positive vs. negative. In the NEWTON pipeline, automatically generated questions are tagged with the polarity of the question, and the corresponding position of the correct answer. This design facet enables users of NEWTON to systematically examine the effects of polarity and position bias for any benchmark generated using NEWTON. Recognizing that mitigation of such biases is an open area of research within NLP, we hope that NEWTON can contribute to the evaluation of strategies aimed at addressing these biases.
>
> > *"The captions are too small. Text in Figure 3 is hard to see. The data statistics in figure 4 and 6 are hard to see."*
>
> Thanks for these valuable suggestions! We have updated our paper to reflect the changes, which will be made available upon final notification of results, per EMNLP policy.
>
> > *Note on Reproducibility*
>
> We noticed that the reviewer indicated a 2 for reproducibility. We have tried our best to make our submission as reproducible as possible by including the source code, dataset, and a detailed outline of the model parameters, validation performance, and evaluation metrics in the main paper. We welcome any additional suggestions to improving the reproducibility of our work!

---

### Official Review · Reviewer_kDhf · 2023-08-08

**Typos Grammar Style And Presentation Improvements:** N/A
**Soundness:** 5

**Excitement:**

4: Strong: This paper deepens the understanding of some phenomenon or lowers the barriers to an existing research direction.

**Missing References:**

N/A

**Paper Topic And Main Contributions:**

This paper introduces NEWTON, a new benchmark of 160K templated question/answers for investigating fine-grained physical reasoning abilities (ability to recognize properties such as texture, brittleness, along with deeper relative reasoning — “I am packing a grocery bag; which object should I put at the bottom?”). Notably, NEWTON also releases a data repository consisting of a curated collection of 700 objects and associated (cleaned) physical attributes, along with a crowdsourcing framework for sourcing more data. Finally NEWTON introduces a custom pipeline that enables new users to synthesize personalized template questions, to quickly generate custom Q/A sets for bespoke dataset/object/attribute splits.

In addition to the resource contribution, the paper evaluates many existing large-scale LLMs on the NEWTON QA benchmark, with careful analysis and thorough findings on the types of physical reasoning current LLMs are good at, and more importantly, where they have room to improve.

**Questions For The Authors:**

A. While I appreciate the NEWTON data repository and benchmark QA task, it’s not immediately clear to me the value of the “NEWTON Pipeline” for synthesizing more bespoke challenge sets. From the paper, there are no examples of what this process would look like — could you speak more to how you would hope this dataset get used?

B. Can you confirm that the crowd-sourcing component described in Section 2.2 will be released along with the rest of the framework?


**Reasons To Accept:**

This is the gold standard for a resource paper on a timely and important topic! As folks in the robotics and vision communities are turning more and more to using LLMs for things like task planning, questions of physical reasoning over fine-grained attributes (beyond existing benchmarks like PIQA) has become increasingly necessary, and this work does an incredible job filling that void, and doing so in a way that can be easily extended by the community.

Furthermore, the existing evaluation on the initial NEWTON QA benchmark is already illuminating, and the findings in just running the baselines will have immediate impact to the community.

**Reasons To Reject:**

I think this is a fresh and useful contribution to the field, so I’m definitely not advocating for rejection in any way; the following is a small weakness/observation that I’m hoping the authors can address to help ensure the longevity of this resource!

A large part of physical reasoning is a notion of grounding to some externalized environment state; while the questions in NEWTON are targeting text-only physical reasoning, I could imagine different conditions objects could be in (e.g., a torn paper bag, a backpack in disrepair) that affect how one might reason over physical properties — a feature the current NEWTON framework seems to lack. Is there a way to extend the benchmark to adding reasoning conditioned on richer object descriptions? Is this already supported?


**Reproducibility:**

5: Could easily reproduce the results.

**Reviewer Confidence:**

4: Quite sure. I tried to check the important points carefully. It's unlikely, though conceivable, that I missed something that should affect my ratings.

---

> ### Author Rebuttal · Authors · 2023-08-28
>
> We thank the reviewer for the positive comments and helpful feedback. Here are our responses addressing the questions and comments that were raised:
>
> > *"A large part of physical reasoning is a notion of grounding to some externalized environment state; while the questions in NEWTON are targeting text-only physical reasoning, I could imagine different conditions objects could be in (e.g., a torn paper bag, a backpack in disrepair) that affect how one might reason over physical properties — a feature the current NEWTON framework seems to lack. Is there a way to extend the benchmark to adding reasoning conditioned on richer object descriptions? Is this already supported?"*
>
> We thank the reviewer for pointing this out, and concur that different conditions that objects could be in affects reasoning over physical properties. In order to encompass these diverse object conditions, it becomes necessary to incorporate perception into the process. This entails representing the object’s state through mediums like images or point clouds. NEWTON currently contains thumbnails and 3D models paired with each category. These visual references were at the disposal of annotators during the crowdsourcing phase, thereby grounding their responses to a certain extent in the depictions of objects. To support more diverse conditions outside of the distribution of objects provided, the authors are actively working on an extension which enables part-level annotations that are grounded on perception, where, for instance, annotators can identify the location of a tear on a paper bag, etc.
>
> > *"While I appreciate the NEWTON data repository and benchmark QA task, it’s not immediately clear to me the value of the “NEWTON Pipeline” for synthesizing more bespoke challenge sets. From the paper, there are no examples of what this process would look like — could you speak more to how you would hope this dataset gets used?"*
>
> We thank the reviewer for pointing this out. We have taken the feedback into account and added an additional section with an example of synthesis of additional challenge sets in the Appendix (the updated version will be made available upon notification of results, per EMNLP policy).
>
> The synthesis process begins with Context and Attribute Specification, where users identify a context, and relevant attributes. Next, object filtering involves using the identified attribute(s) to *automatically* filter objects into a positive set and a negative set. Using the grouped objects, one can specify query templates, and automatically populate the templates to synthesize a diverse and customized challenge set. Using this, language models can be evaluated for accuracy in the specific user-identified context to find the optimal prompting strategy and model. Given the task of packing purchased items into a box at a department store, our objective is to evaluate the language models’ ability to factor in the attribute of stiffness. To achieve this, we create a novel challenge set that aligns with the specific requirements and focus of the task. Since we have access to a large repository of object-attribute pairs, we can effortlessly generate nearly infinite combinations to populate the question template. This empowers users to harness the NEWTON repository of object-attribute pairs to generate questions which suit their own application. By doing so, they can assess the language models’ competence in addressing the task and identify the most suitable language model for their needs.
>
> > *"Can you confirm that the crowd-sourcing component described in Section 2.2 will be released along with the rest of the framework?"*
>
> Yes, we aim to open-source the interface for the crowd-sourcing component.

---

### Meta-Review · Area_Chair_g3YH · 2023-09-17

**Recommendation:** 4

**Metareview:**

The paper presents "NEWTON", a benchmark designed to evaluate fine-grained physical reasoning abilities in Large Language Models (LLMs). NEWTON comprises 160K templated question/answers that focus on properties such as texture and brittleness, as well as deeper relative reasoning scenarios. The benchmark is built using a curated collection of 700 objects with associated physical attributes. A unique feature of NEWTON is its custom pipeline that allows users to synthesize personalized template questions for custom dataset/object/attribute splits. The paper also evaluates several LLMs on the NEWTON QA benchmark, providing insights into their strengths and weaknesses in physical reasoning. The benchmark aims to fill the gap in evaluating LLMs' ability to reason about physical properties and scenarios.

NEWTON is a timely and valuable contribution to the field, addressing the increasing need for benchmarks that evaluate LLMs' physical reasoning abilities, especially as robotics and vision communities increasingly use LLMs. The resource is extensive, covering a wide range of objects and physical attributes, making it significantly larger than existing benchmarks. The paper is well-structured, with clear motivations and comprehensive experiments. Ablation studies further strengthen the paper's arguments, and the fine-tuning of models on the dataset is a commendable effort. Overall, the paper provides a systematic approach to studying physical reasoning, making it a significant addition to the community.

While NEWTON is a valuable contribution, it has some limitations. The benchmark might lack the ability to reason based on richer object descriptions, such as the condition of objects, which can affect physical reasoning. Questions can be ambiguous due to the general nature of objects, as objects made of different materials can have varied physical properties. The authors, during the rebuttal, provided the reviewers with effective answers and new relevant experiments.

---

### Decision · Program_Chairs · 2023-10-07

**Decision:**

Accept-Findings

**Comment:**

The paper presents "NEWTON", a benchmark designed to evaluate fine-grained physical reasoning abilities in Large Language Models (LLMs). NEWTON comprises 160K templated question/answers that focus on properties such as texture and brittleness, as well as deeper relative reasoning scenarios. The benchmark is built using a curated collection of 700 objects with associated physical attributes. A unique feature of NEWTON is its custom pipeline that allows users to synthesize personalized template questions for custom dataset/object/attribute splits. The paper also evaluates several LLMs on the NEWTON QA benchmark, providing insights into their strengths and weaknesses in physical reasoning. The benchmark aims to fill the gap in evaluating LLMs' ability to reason about physical properties and scenarios.

NEWTON is a timely and valuable contribution to the field, addressing the increasing need for benchmarks that evaluate LLMs' physical reasoning abilities, especially as robotics and vision communities increasingly use LLMs. The resource is extensive, covering a wide range of objects and physical attributes, making it significantly larger than existing benchmarks. The paper is well-structured, with clear motivations and comprehensive experiments. Ablation studies further strengthen the paper's arguments, and the fine-tuning of models on the dataset is a commendable effort. Overall, the paper provides a systematic approach to studying physical reasoning, making it a significant addition to the community.

While NEWTON is a valuable contribution, it has some limitations. The benchmark might lack the ability to reason based on richer object descriptions, such as the condition of objects, which can affect physical reasoning. Questions can be ambiguous due to the general nature of objects, as objects made of different materials can have varied physical properties. The authors, during the rebuttal, provided the reviewers with effective answers and new relevant experiments.